

# Linking local vulnerability assessments to climatic hazard losses for river basin management

Hung-Chih Hung[1], Yi-Chung Liu[2], Sung-Ying Chien[2]

[1] Department of Real Estate and Built Environment, National Taipei University, New Taipei City, 23741, Taiwan

[2] National Science and Technology Center for Disaster Reduction, New Taipei City, 23143, Taiwan

*Correspondence to*: Hung-Chih Hung (hung@mail.ntpu.edu.tw)

**Abstract.** To prepare for the potential impact of climate change and related hazards, many countries have implemented integrated river basin management programs. This has led to significant challenges for local authorities to improve their understanding of how the vulnerability factors are linked to losses in climatic disaster. This article aims to examine whether highly vulnerable areas experience significantly more damage at the river basin levels due to weather extreme events, and investigates the vulnerability and hazard impact factors determine losses in a disaster. Using three river basins in southern Taiwan that were seriously affected by Typhoon Morakot in 2009 as case studies, a novel methodology is proposed that combines a geographical information system (GIS) and a multicriteria decision analysis (MCDA) to evaluate and map composite vulnerability to climatic hazards across river basins. The linkages between the hazard impacts, vulnerability factors and disaster losses are then tested using a disaster damage model (DDM). The results of the vulnerability assessments demonstrate that almost all of the most vulnerable areas are situated in the regions of the middle, and upper reaches and some coastlines of the river basins. The losses and casualties due to typhoon are significantly affected by local vulnerability contexts and hazard impact factors. Finally, policy implications to minimize vulnerability and risk and for integrated river basin governance are suggested.

## 1 Introduction

Major portions of Asia have an increasing exposure and vulnerability to climate change and weather extremes due to rapid urbanization and overdevelopment in hazard-prone areas (IPCC, 2014). In August 2009, a devastating typhoon (Morakot) hit three major river basins in southern Taiwan. Approximately 700 people were killed and total economic losses were estimated to have been US$ 0.6 billion (Liu et al., 2014). Therefore, integrated river basin management (RBM) programs that can cope with and reduce the potential impacts of climate change and climate-related (climatic) disaster risks have become more

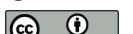


important (Hung et al., 2013).

Integrated water resource management is a process that promotes the coordinated development and management of water, land uses and related resources (GWP 2000). An integrated RBM program adopts the river basin as a management unit, and has a comprehensive perspective that connects water resource management, agricultural irrigation and land use planning to

5    build more resilient river basin contexts (Penning-Rowsell et al., 2006). Vulnerability assessment plays a vital role for decision makers in scrutinizing the biophysical, socioeconomic conditions and their distributions over river basins. This process of assessment also allows decision makers to integrate various local connections into planning and policy lines to mitigate disaster damage and risk within the context of whole river basins (Hooijer et al., 2004; Hung and Chen, 2013; You and Zhang, 2015).

10    Vulnerability analyses have focused more on assessing, mapping and distinguishing the variability of the vulnerability distribution between regions (Adger, 2006; Hung and Chen, 2013; Ahumada-Cervantes et al., 2015). However, climatic disaster losses and risk accumulation result from the interlinks between hazard impacts, exposure and vulnerability components (UNISDR, 2012; Hung et al., 2013). The majority of extant studies infer disaster losses and risks using computer-aided simulation, scenario analyses and multicriteria decision analysis (MCDA) (Tate et al., 2010; Ni et al., 2010;

15    Hung et al., 2013; De Bruijn et al., 2014). The findings allow disaster risk, impact and distributions to be characterized and enable decision makers to create risk maps and to communicate high risk areas with stakeholders. Few studies have systematically examined how the vulnerability and hazard impact factors are linked to their potential effect on disaster losses. This compromises the application of existing vulnerability and exposure studies to disaster risk assessment and integrated RBM.

20    This article aims to examine whether geographic localities that are characterized by high vulnerability experience significantly more damage at the river basin level due to extreme weather events, and identifies the vulnerability, hazard and exposure factors that influence these damage or losses. Using three river basins in southern Taiwan that were affected by Typhoon Morakot as case study areas, we propose a novel methodology that is based on existing disaster impact theory, which combines an MCDA, GIS (geographical information system)-based statistics with a multivariate analysis to assess

25    vulnerability to climatic hazard (especially typhoons and floods). Moreover, the connection between vulnerability, hazard impact factors and disaster losses is then examined using a disaster damage model (DDM). This methodology may also be



applicable to other river basins. Finally, we suggest the policy options to build adaptive capacity and improve RBM using the findings in this study.

## 2 Vulnerability and disaster impacts

### 2.1 Vulnerability assessment

5      Vulnerability assessment has been broadly used in various fields that are related to climate change adaptation and disaster risk management, although there is no common view on the concept of vulnerability. In disaster impact research, vulnerability is widely described as the degrees of susceptibility of these assets to damage and loss (UNISDR, 2013). IPCC (2014) conceptualized vulnerability as encompassing a variety of concepts and elements, including sensitivity or susceptibility to harm and lack of capacity to cope and adapt.

10      Watershed contexts consist of various biophysical, socioeconomic, industrial and land use elements. Therefore, from the perspective of integrated RBM, vulnerability assessment should facilitate decision-makers to engage in an integrated analysis of the interaction between the components of vulnerability and the properties of a specific watershed context (O'Brien et al., 2007; Engle and Lemos, 2010; Hung and Chen, 2013). A targeted integrated RBM should integrate IPCC's (2014) with UNISDR's (2013) concepts to build a more transdisciplinary and comprehensive vulnerability assessment

15      framework. Therefore, vulnerability can be generally described as a function of exposure, sensitivity and adaptive capacity:

$$\text{Vulnerability} = f \text{ (exposure, sensitivity, adaptive capacity)} \qquad (1)$$

### 2.2 Vulnerability and disaster losses

Existing approaches on the investigation of the relationships between vulnerability and disaster losses can be divided into two major types. The first type interprets disaster damage or risk as a function of vulnerability, and frequently uses

20      catastrophic, PSR (pressure-state-response), PAR (pressure-and-release) theories or MCDA with computer-aid simulation and GIS-based analysis to predict disaster losses (Ermoliev et al., 2000; Wisner et al., 2004; Tate et al., 2010; Scheuer et al., 2011; De Bruijn et al., 2014). Therefore, disaster risk or potential losses can be directly predicted by a vulnerability assessment (Cutter et al., 2003; Hung and Chen, 2013). This type of research uses a top-down approach that can bring the disaster information that is related to predicted distributions of disaster impacts and risk to the fore, although there are



uncertainties and ambiguities in the processes of projection.

The second type focuses on bottom-up and data-based analysis, which often uses historical or survey data to characterize the disaster damage (Zahran et al., 2008; Bhattarai et al., 2015). This approach concentrates more on mapping the disaster damage distribution at national or regional levels. It also mostly combines expert judgment with mono-dimensional evaluation to identify the factors that cause disaster damage (Mokrech et al., 2012; Hung and Chen, 2013). The results not only allow decision makers to identify disaster loss distributions, but also enhance the understanding of their determinants (Downton and Pielke, 2005). However, there is relatively little linking of a multi-dimensional vulnerability assessment with an empirically-based disaster loss evaluation in the context of river basins.

Using a theoretical simulation and MCDA, the first type approach seeks to systematically identify disaster losses and scrutinize their components, and to predict the impacts of the various disasters that result from different hypothetical events. In contrast, the second type approach enables decision makers to conjointly treat both quantitative disaster loss data analysis and qualitative human judgment. Both types of approaches consider disaster losses as both inherent and dynamic because of the ongoing interaction between the impacts of climatic hazards and the biophysical and socioeconomic dimensions of vulnerability in a watershed system (O'Brien et al., 2007; Maru et al., 2014).

Increasing the understanding of the evolution of climatic disaster risk highlights the importance of connecting these two approaches and their relative magnitudes (Mokrech et al., 2012; Visser et al., 2014). Incorporating the first type into the second type approach allows the frameworks to be created for disaster risk analysis that might extend the range of vulnerability assessments and allow them to be sequenced to generate robust resilience and adaptation pathways (Hung et al., 2016).

**3 Methods and data**

To characterize the disaster loss distributions and their linkages with various vulnerability components, an MCDA and GIS-based statistic analysis are integrated with a data-based multivariate analysis. First, the composite vulnerability framework is constructed to summarize a review of the literature and combined with an MCDA to assess climatic hazard vulnerability at the river basin level. Second, using the DDM, the relationship between disaster loss distributions, impacts and vulnerability factors is tested and compared using numerous regression models. Finally, the results are discussed and

implications for better adaptation policies are highlighted.

### 3.1 Indicators for the vulnerability framework and hypotheses

The assessment framework created here is based on the IPCC's (2014), UNISDR's (2013) concepts of vulnerability and literature review. This framework takes advantage of the contributions of existing knowledge and embraces the synergies and complexities of watershed contexts, as discussed in detail in Hung and Chen (2013). The indicators involved in the framework consist of three dimensions: exposure, sensitivity and adaptive capacity. To assess the integrated vulnerability, we use the framework of vulnerability indicators that was proposed by Hung and Chen (2013), which is appropriate and is widely applied to the river basin conditions in Taiwan. An assessment of composite vulnerability is then conducted across the case study areas at the village scale, which is the basic unit of local administration in Taiwan.

### 3.1.1 Exposure indicators

Exposure refers to the biophysical factors and the extent to which the properties of vulnerable system are in contact with hazards (Hung et al., 2016). To reflect the degrees of exposure, *averaged annual rainfall* and *potential debris flow torrents* are used. The expectation is that either higher rainfall or greater debris flow torrents enhance vulnerability and increase the levels of disaster losses (Wisner et al., 2004).

### 3.1.2 Sensitivity indicators

Sensitivity is a widely used attribute in climate change and disaster risk management to describe vulnerability (Cutter et al., 2003; O'Brien et al., 2014). The sensitivity indicators are mostly composed of inherent biophysical and societal contexts. The societal context can be further classified into socioeconomic and land use sensitivity (Hung and Chen, 2013).

The hypothesized links between the biophysical context and disaster losses are defined in the effect of *proximity to rivers* and *elevation* indicators on disaster losses. The areas where there is greater proximity to rivers and/or higher levels of elevations are both more vulnerable and more sensitive to disaster damage (Ni et al., 2010). The socioeconomic indicators include *populations*, *social dependence*, *income*, *employment* and *production values* of industries and services. These indicators reflect the contextual vulnerability and fragility in a watershed. Therefore, increasing income, employment and/or production values is expected to enhance coping strategies, and to decrease vulnerability and potential losses in disaster

(Zahran et al., 2008). In contrast, populations and social dependence are expected to have a positive relationship with disaster damage (Hung et al., 2016).

In terms of land-use, the indicators comprise *urban developments*, *agricultural uses*, *envornmental sensitive areas* and *road infrastructures*. Generally, while preserving more sensitive areas decreases vulnerability and disaster losses, greater urban development, agricultural use or road infrastructure would increase land use density, agricultural development and tourist activity, which results in higher vulnerability and greater expected disaster losses (Cutter et al., 2003; Mehaffey et al., 2008).

### 3.1.3 Adaptive capacity indicators

Adaptive capacity indicators are used to measure the ability of communities to adjust to potential damage, to take advantage of opportunities, or to respond to disaster consequences (IPCC, 2014). The indicators include *shelters*, *medical*, *fire and police* services. These indicators represent an area's abilities of coping, evacuation and emergency responses. Therefore, improving these facilities reduces vulnerability and the likelihood of disaster damage. Behavior and heuristic factors, such as residents' *risk perceptions*, the ability to *access resources* and to successfully adapt to hazards (self-efficacy) are also considered. The hypothetical relationships between these factors and disaster damage are negative (Eakin et al., 2010). Finally, the indicators that are used to assess vulnerability are shown in Table 1, along with their descriptions, data sources and the expected direction of the relationship to disaster losses.

(**Table 1.** Hazard impacts, vulnerability indicators (variables) and expected sign to disaster losses)

### 3.2 Composite vulnerability index

The composite vulnerability index (CVI) is used to assess the integrated vulnerability for each village in the river basins. However, the survey values of various indicators contain different scales and units. A min-max scaling is used to directly normalize all of the data into a uniform [0, 1] scale with ratio properties. The normalized values are then used to compute CVIs as:

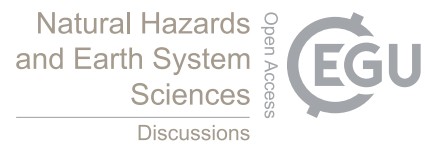

$$\text{CVI}_i = \sum_{j=1}^{m} w_j x_{ij.} \hspace{3cm} (2)$$

where $\text{CVI}_i$ represents the composite vulnerability index for village $i$; $x_{ij}$ denotes the normalized value for indicator $j$ and $w_j$ is the weight. If $x_{ij} > 0$, levels of overall vulnerability are greater; if $x_{ij} < 0$, the overall vulnerability is decreased. Equal-weights are assigned to each indicator in order to give an equivalent basis on which the attributes of vulnerability and disaster losses in the river basins can be compared.

### 3.3 Linking vulnerability factors and climatic disaster losses

This study focuses on single-scenario disaster event to enable comparative static modeling of damages and losses at different points over river basins. This approach allows disaster scenarios to be controlled, and which, other things being equal, any variation in losses directly are resulted from changes in hazard impacts and vulnerability factors (Hung et al., 2013). Therefore, the disaster damage model can be written as the following function:

$$\text{Disaster loss} = f\,(\text{hazard, vulnerability}) \hspace{2.5cm} (3)$$

Equation (3) implies that the interaction of hazard impact and vulnerability generates disaster losses. Therefore, the extent of disaster damage and/or losses varies with vulnerability contexts and the impact of climatic hazards (i.e., typhoon events), while hazard impact is often deemed as outerdependant factors. To more specifically identify the relationship between disaster losses and vulnerability factors, several regression models are used in the case study.

### 3.4 Case study areas and data

This study explores three very different river basins in southern Taiwan, with various degrees of development and different contexts (Fig. 2). All were significantly affected by Typhoon Morakot in 2009. The case study areas include three major river basins: Gaoping, Tsengwen and Taimali River. According to the 2015 census, these three river basins encompass 598 villages, around 1.26 million populations and cover an area of approximately 7,885km$^2$. Highly diversified topography is distributed over these three watersheds. The altitude of this region ranges from coastal lowlands along the western shoreline to above 3,000 meters in the eastern mountainous areas. Uncontrolled urban sprawl and environmental destruction combined with the growing threat of climate change and weather extremes mean that these are some of the riskiest regions in Taiwan

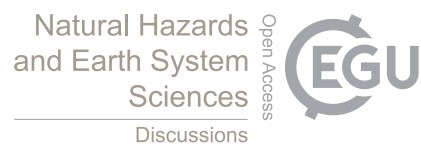

(Liu et al., 2013).

The data that were used to model the linkages between vulnerability factors, disaster impacts and losses were collected from multiple sources. The disaster loss database for Typhoon Morakot was systematically collected by the Department of Science and Technology, Taiwan. This database included the surveyed numbers of casualties, property and agricultural losses, the distributions of inundation and landslides and damage to public facilities. The data on vulnerability factors came from official censuses and random sampling face-to-face questionnaire surveys of residents (shown in Table 1 in detail).

(**Figure 1.** Distributions of the estimated composite vulnerability indices over three river basins)

## 4 Results and discussions

### 4.1 Composite vulnerability assessments

Using the estimated CVIs, Fig. 1 shows the distributions of the estimated index values superimposed on the geopolitical boundaries of villages throughout the three river basins. The CVI estimates were divided into five levels (at 20% intervals). The villages with estimated index values within the 80-100[th] percentiles were defined as the most vulnerable, and those within the 1[st]-20[th] percentiles as the least vulnerable.

Fig. 1 shows that there is a high degree of heterogeneity in the spatial distributions of the estimated composite vulnerability across the study areas. In the Tsengwen River basin, the most vulnerable areas were concentrated in the middle reaches and along some coastlines. Most of the middle and upper reaches of the Gaoping and Taimali River basin (especially northern shore) had the most vulnerable villages, and most of the lower reaches had the least vulnerable ones. These spatial distribution patterns conform well with the historical experience of numeral typhoons in these areas that had resulted in serious casualties, property and crop losses.

The results corroborate similar findings from related studies (Hung and Chen, 2013; Liu et al., 2013) and significantly show that specifically-defined clusters of highly vulnerable areas are mostly situated in midstream and upstream reaches. This presents a challenge for watershed managers to understand why these areas are particularly vulnerable and how they are linked to disaster losses, as well as what the appropriate range of adaptation and policy options are identified to reduce risk.



### 4.2 The distributions of losses due to Typhoon Morakot

The inundation areas due to Typhoon Morakot were concentrated in the regions where the Gaoping River converges with its tributaries, and the major landslide and debris flow torrents occurred in the middle and upper reaches. This affected the distributions of property, public facility and agricultural damage (Fig. 2). Using a $t$ test for correlation analysis, the location of agricultural damage significantly corresponds to the positions of the landslides (Spearman $\rho= 0.18$, $p< 0.01$; Pearson $r= 0.43$, $p< 0.01$) and damaged bridges. The pattern of casualties also highly correlated with the numbers of landslides (Spearman $\rho= 0.22$, $p< 0.01$, Pearson $r= 0.23$, $p< 0.01$) and damaged bridges (Spearman $\rho= 0.40$, $p< 0.01$, Pearson $r= 0.42$, $p< 0.01$).

(**Figure 2.** Distributions of the losses due to Typhoon Morakot over three river basins)

In the Tsengwen river basin, flooding and landslides caused more serious damage to the watersheds than debris flow torrents. This resulted in both casualty counts and agricultural losses that were significantly associated with the patterns of landslides (casualties: Spearman $\rho= 0.17$, $p< 0.05$, Pearson $r= 0.53$, $p< 0.01$; agriculture: Pearson $r= 0.56$, $p< 0.01$) and damaged bridges (casualties: Spearman $\rho= 0.27$, $p< 0.01$, Pearson $r= 0.55$, $p< 0.01$; agriculture: Pearson $r= 0.40$, $p< 0.01$). Agricultural and property losses in the Taimali watershed were mostly concentrated along the road systems, so there was a noteworthy relationship between road infrastructure, land development and disaster loss that requires further investigation.

### 4.3 The determinants of disaster losses

The regression analyses that were used to identify the determinants of typhoon losses included casualties, property and agricultural losses. The choice of regression models depends on the type of distribution for the disaster loss data. The distribution of disaster casualties is non-normal. Zero counts significantly skew the distribution leftward– 93% of villages Typhoon Morakot caused no recorded injuries or fatalities. The total casualties were 684. The arithmetic mean is 1.01 and the standard deviation is 18.76, and dispersion is 18.6 times greater than the average. The casualties were a non-negative integer that exhibit significant over-dispersion with a disproportionate number of zero counts. The data were thus investigated using a ZINB (zero-inflated negative binomial) or ZIP (zero-inflated Posisson) regression model, which allows



us to estimate the net effects of independent vulnerability factors on casualties (Cameron and Trivedi, 1998; Zahran et al., 2008). To more comprehensively scrutinize the effect of disaster losses, the integrated typhoon loss index (ITLI) was estimated and acts as a proxy for combined losses due to the typhoon:

$$ITLI_i = Agiculture_i + Property_i + Casualty_i. \qquad (4)$$

where $Agiculture_i$ and $Property_i$ are agricultural and property losses for village $i$, respectively, and $Casualty_i$ is casualty counts. A Lagrange multiplier (LM) test points to which the ITLI is a non-negative rational number that spreads in a certain range. Therefore, we applied a Tobit (Censored) regression model to examine the affecting factors for the ITLI.

(**Table 2**. Regression analyses of the determinants of climatic disaster losses)

Table 2 reports the results of the ZINB and ZIP regression analyses for typhoon casualties, and the Tobit models for the

ITLIs. Six separate models were estimated, with predictors for each watershed (excluding Taimali River due to little sample size) and for all three river basins. To screen variables for multicollinearity, we used zero-order correlation and Variance Inflation Factor tests in an Ordinary Least Squares regression. This showed that the *risk perceptions* and *access to resources* had a significantly high multicollinearity with other variables, so these two variables were eliminated in some regression analyses.

For all of the regression models, the results indicated that most hazard impact factors played an important role in determining typhoon casualties and losses. As expected, landslides, damaged bridges, agricultural losses, property losses and flooding were positively associated with typhoon losses, although agricultural losses were negatively related to casualty counts in the Gaoping watershed. These findings correspond with the PSR framework that considers the hazards as pressures, so their impact would change the quality of the environment. The greater the hazard impact, the greater was the probability

of casualties and disaster losses (OECD, 1993; Wisner et al., 2014).

In terms of the biophysical exposure indicators, average rainfall was a major positive contributor to the casualty counts in both the Gaoping and the Tsengwen watersheds, while it was a negative predictor of disaster losses. In the Gaoping River basin, the highest numbers of casualties occurred in the areas with higher levels of rainfall and elevations rather than in areas where there were debris flow torrents. The areas within 0-200m of rivers exhibited significantly increased numbers of

casualties in all three river basins, and increased typhoon losses in both the Gaoping and the Tsengwen watersheds. Most of these results are consistent with expectations and the results of earlier studies on the linkage between biophysical factors and disaster losses (OECD, 2012; Hung et al., 2016). It implies that areas with a higher risk were mostly located in regions with higher elevations and greater proximity to the rivers over the three watersheds.

In the compilation of socioeconomic factors, population density was a strong predictor of casualty counts and disaster

losses and was negatively related to casualty counts, but its relationship to disaster losses was positive (except for the Gaoping River basin). The results reflect that the patterns of disaster damage depend on the type of hazard impact. The upstream areas frequently had a low population density, but more landslides occurred, so there were more casualties. Generally, inundation mostly occurred in downstream areas, which led to more overall losses than casualties.

The lower income areas were likely generating more casualties. As the production values of industries and services in an

area increased, there was an increase in the capacity for pre-disaster preparedness and emergency responses, which decreased disaster loss and risk. With the exception of a significantly positive relationship between employment rates and casualty numbers, and between social dependence counts and disaster losses in the Gaoping watershed, the other socioeconomic factors had a weak relationship with casualty and loss distributions. These results do not fully conform with the results of existing studies, which claim that the relationship between social vulnerability factors and disaster losses (or

risk) is functional (Zahran et al., 2008; Hung and Chen 2013). Rather, their relationships remain complex and difficult to model, and depend on multiple effects of local contexts and disaster impacts for each watershed (UNISDR, 2012).

For all of the regression models, increasing urban or agricultural development significantly increased casualty counts and typhoon losses, although increasing agricultural uses strongly decreased casualty distributions in the Gaoping watershed. These results also showed that more sensitive areas exhibited a decreased occurrence of casualties and losses (except for the Tsengwen watershed). Road or transportation infrastructures would be helpful in the evacuation and disaster relief, which led

to fewer casualty counts after typhoon hitting. As most studies emphasized (Mehaffey et al., 2008; Hung et al., 2016), this study shows that higher levels of urbanization and farming reclamation increase vulnerability to hazard and result in higher damage.

The adaptive capacity variables also played a critical role in predicting disaster damage. Especially, an increase in medical services, access to resources and self-efficacy significantly attenuated losses in a disaster, and resulted in strongly decreased

casualty counts in both the Gaoping and the Tsengwen river basins. These results confirm earlier findings that improvements in adaptive capacity effectively reduce disaster damage and risk (Eakin et al., 2010; Hung and Chen, 2013). However, one noteworthy exception was that the areas with a greater ability to access resources experienced more typhoon casualties in all three river basins. One possible explanation is that most of these areas are particularly vulnerable and frequently receive large amounts of external aid in the aftermath of a disaster. This aid probably only provides temporary disaster relief and

does not prevent long-term vulnerability.

**4.4 Policy implications**

This study is an initial analysis of the relationship between vulnerability attributes, hazard impact and disaster loss. Using composite vulnerability assessments and regression analysis, it shows that villages with higher elevations, in upper streams and with greater proximity to rivers tend to suffer more casualties and losses in a disaster because of their greater exposure to

20 typhoon impacts. However, constraints associated with local government adaptation in the river basins highlight a range of challenges in relation to the way in which an integrated RBM adaptation is structured. Efforts to facilitate adaptation should target the mitigation of vulnerability and risk by combining resilient infrastructure, warning systems and risk communication

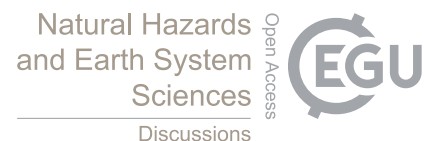

to improve the emergency systems to allow pre-disaster hazard-mitigation planning to reduce risk and save the lives (Hung and Chen, 2013; Hung et al., 2013).

Long-term policy for integrated RBM should take account of land use planning and regulations, relocation and building codes to prevent urban and agricultural developments from encroaching into hazard-prone areas (Neuvel and van den Brink, 2009). The vulnerability distributions and their linkages to disaster losses as presented in this study will enable policy makers to generate hazard risk maps that identify the riskiest areas and communicate this information to stakeholders. In the upper streams, land use management can be further integrated into river basin governance in order to keep the environmental sensitive areas from excessive urban sprawl, agriculture and tourism activities and to appraise adaptation options for the most vulnerable areas. Besides structural engineering projects, downstream areas need to incorporate wetland preservation, flood insurance, warning systems and related risk-sharing arrangements into the existing RBM framework to minimize risk.

**5 Conclusions**

Increasing climate change and the effects of weather extremes pose imminent challenges and high uncertainties for the RBM. Therefore, an understanding of the links between disaster impact, vulnerability factors and disaster losses is critical for hazard risk and river basin governance and the evaluation of adaptation strategies.

This article proposes a novel approach that combines elements of previous studies on vulnerability assessments and disaster impacts to characterize the vulnerability over river basins and to examine its influence on typhoon losses. A composite vulnerability assessment framework is constructed in hybridized with an MCDA to create vulnerability maps that inform policy-making and identify the core areas in which adaptive measures are most needed to reduce vulnerability and risk. Various regression models are used to examine the key vulnerability and hazard impact factors that determine the casualties and losses caused by Typhoon Morakot, and to compare the typhoon losses for three river basins in terms of the variability in local contexts.

The findings indicate that both the hazard impacts and the vulnerability factors have a strong relationship to the spatial distribution patterns of disaster losses. In particular, local biophysical, socioeconomic and land use attributes are key predictors to disaster losses. Local agencies have to make some tradeoffs between building adaptive capacity and reducing vulnerability. However, the disaster event considered in this study is limited. Further case studies across other river basins



will provide greater insights into the effects of the tradeoffs in terms of reducing risk. The robustness and application of our modeling method can also be verified by comparing the operationalized loss surveys for other cases in the aftermaths of other disaster events. This study underscores useful policy and land use planning implications for an integrated RBM that ensures more resilient river basins.

**Acknowledgements**

The authors would like to acknowledge the assistance from the Ministry of Science and Technology, Taiwan under Grants: MOST 103-2625-M-305-001. None of the conclusions expressed here necessarily reflect views other than those of the authors.

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

**Figure 1.** Distributions of the estimated composite vulnerability indices over three river basins





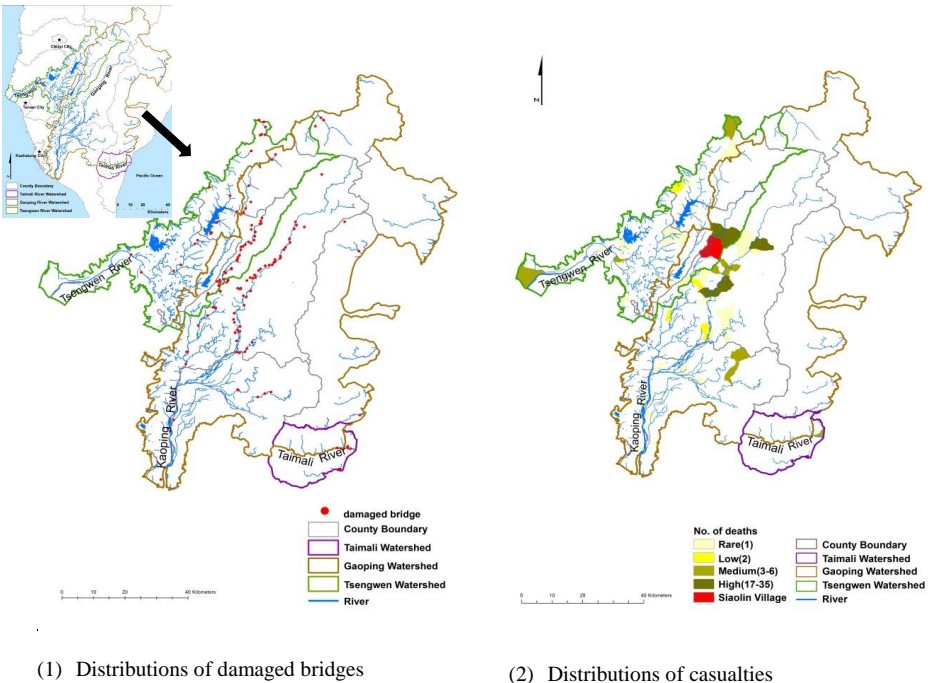

(1) Distributions of damaged bridges

(2) Distributions of casualties

**Figure 2.** Distributions of the losses due to Typhoon Morakot over three river basins





**Table 1.** Hazard impacts, vulnerability indicators (variables) and expected sign to disaster losses

| Category | | Indicator | Description | Data source | Mean (S.D.) | Sign |
|---|---|---|---|---|---|---|
| **Hazard impacts** | | Casualties | Number of casualties (people) | NCDR[a], Taiwan | 1.01 (18.76) | + |
| | | Landslides | Areas of landslides (km$^2$) | NCDR, Taiwan | 0.48 (2.22) | + |
| | | Damaged bridges | Number of damaged bridges | NCDR, Taiwan | 0.24 (0.90) | + |
| | | Agricultural losses | Amount of agricultural losses (1000 NT$) | NCDR, Taiwan | 14.01 (39.34) | + |
| | | Property losses | Number of damaged dwelling | NCDR, Taiwan | 48.68 (126.8) | + |
| | | Flooding areas | Areas of inundation (km$^2$) | NCDR, Taiwan | 0.32 (0.94) | + |
| **Exposure** | | Rainfall | Averaged annual rainfall (mm) | Central Weather Bureau, Taiwan | 1932 (364) | + |
| | | Debris flow torrents | Number of potential debris flow torrents and landslides | Council of Agriculture, Taiwan | 0.41 (1.07) | + |
| **Sensitivity** | Biophysical context | Proximity to rivers | Areas within 0m-200m to rivers (km$^2$) | Measured by GIS | 0.18 (0.21) | + |
| | | Elevation | Averaged elevation (m) | Ministry of the Interior, Taiwan | 169.7 (355.3) | + |
| | Socioeconomic sensitivity | Populations | Population density (populations/km$^2$) | Ministry of the Interior, Taiwan | 2.74 (5.60) | + |
| | | Social dependence | Ratio of people over age 65 and under age 6, and females (%) | Ministry of the Interior, Taiwan | 58 (5) | + |
| | | Income | Annual disposable household incomes (1000 NT$) | DGBAST[b], Taiwan | 660.1 (23.7) | − |
| | | Employment | Employed population (employed population/ population) | DGBAST, Taiwan | 0.15 (0.26) | − |
| | | Production values | Annual production values of industries and services (million NT$) | DGBAST, Taiwan | 27.9 (81.4) | − |
| | Land uses | Urban developments | Area of residential, commercial, industrial, educational and public land uses (km$^2$) | Land Use Investigation of Taiwan | 0.35 (0.48) | + |
| | | Agricultural uses | Areas of agricultural land uses (km$^2$) | Land Use Investigation of Taiwan | 2.17 (2.91) | + |
| | | Sensitive areas | Environmental sensitive areas (km$^2$), e.g., flood plain, mountain slope reserve areas | Land Use Investigation of Taiwan | 10.4 (40.4) | − |
| | | Road infrastructure | Areas of road infrastructure (km$^2$) | Ministry of the Interior, Taiwan | 0.16 (0.14) | + |
| **Adaptive capacity** | | Shelters | Number of shelters | Measured by GIS | 1.16 (1.43) | − |
| | | Fire and police services | Number of fire and police manpower | County and city government | 2.05 (2.02) | − |
| | | Medical services | Hospital beds | County and city government | 10.5 (16.0) | − |
| | | Risk perceptions | Average levels of perceived residential risk to climate hazards (5-point Likert scale) | Questionnaire interviews | 2.97 (0.17) | − |
| | | Access to resources | Average levels of ability to access to resources (5-point Likert scale) | Questionnaire interviews | 2.03 (0.18) | − |
| | | Adaptation appraisal | Average levels of residents evaluate their ability to perform adaptations successfully (5-point Likert scale) | Questionnaire interviews | 2.43 (0.50) | − |

[a]: National Science and Technology Center for Disaster Reduction,; [b]: Directorate-General of Budget, Accounting Statistics





**Table 2**. Regression analyses of the determinants of climatic disaster losses

| Variable | All river basins | | Gaoping River basin | | Tsengwen River basin | |
|---|---|---|---|---|---|---|
| | ZINB | Tobit | ZIP | Tobit | ZIP | Tobit |
| Constant | 31.26***(2.98)[a] | -15.46 (-0.82)[a] | 1.98 (0.34) | 72.42**(2.40) | 20.97 (1.63) | -15.04 (-0.62) |
| Landslides | 0.93***(3.08) | 0.23***(4.11) | 0.50***(6.27) | 0.28***(3.83) | -0.53 (-0.47) | 0.12***(7.95) |
| Damaged bridges | 0.61***(4.18) | 6.04***(7.37) | 1.30***(16.37) | 6.57***(6.12) | 0.89**(2.62) | 1.40*(1.65) |
| Agricultural losses | -0.52 (-1.14) | — | -0.80***(-4.62) | — | 0.18*(1.67) | — |
| Property losses | 0.004***(2.57) | — | 0.005***(8.94) | — | 0.009***(2.58) | — |
| Flooding areas | 0.54***(2.70) | 3.36***(4.66) | 1.01***(4.79) | 5.29***(3.81) | 3.28***(2.64) | 6.68***(3.73) |
| Rainfall | -0.001(-0.13) | -0.005**(-2.28) | 0.001***(3.49) | -0.01***(-3.40) | 0.003*(1.74) | -0.003 (−1.07) |
| Debris flow torrents | -0.03 (-0.17) | 0.67 (0.88) | -0.30***(-4.90) | -0.009 (-0.01) | -0.13 (-0.43) | -0.80 (-1.04) |
| Elevation | 0.03***(3.21) | 0.01***(2.86) | 0.003***(9.51) | 0.01**(2.41) | 0.001 (0.62) | -0.003 (-0.79) |
| Proximity to rivers | 0.32*(1.66) | 0.28 (0.70) | -0.69 (-0.58) | 0.12**(2.19) | -0.15 (−0.61) | 0.97**(2.05) |
| Population density | -1.10***(-3.45) | 2.65***(3.36) | 0.86(1.29) | -0.001**(-1.97) | -1.81**(-2.28) | 6.75***(4.59) |
| Social dependence | -5.80 (-0.80) | -5.04 (-0.37) | -2.67 (-0.35) | 30.10*(1.58) | -7.70 (-1.02) | -1.22 (-0.10) |
| Income | -2.59*(-1.74) | 0.0004 (0.15) | -0.007***(-5.17) | 0.002 (0.21) | -0.002 (-0.57) | -0.001 (-0.35) |
| Employment | -0.40 (-0.12) | -1.52 (-0.38) | 4.84**(2.71) | -3.50 (-0.41) | -3.52 (-0.58) | 1.38 (0.45) |
| Urban developments | 0.30***(4.31) | 0.70***(3.82) | 0.24***(10.22) | 0.69***(3.03) | 0.32***(2.52) | 0.40*(1.65) |
| Agricultural uses | 0.12***(2.54) | 3.28***(5.61) | -0.32***(-4.51) | 3.00***(3.28) | 0.12 (1.07) | 0.14***(3.76) |
| Sensitive areas | -0.15***(-3.23) | -0.11***(-3.05) | 0.34***(3.02) | -0.13***(-2.93) | -0.17**(-1.96) | 1.58*(1.73) |
| Road infrastructure | -1.52***(-2.99) | 0.17 (0.78) | -1.11***(-4.09) | -1.85 (-1.06) | -1.10*(-1.66) | -0.71 (-0.89) |
| Production values | -0.12 (-0.55) | -0.18*(-1.66) | -0.49***(-3.53) | -0.35 (-0.18) | 0.76 (0.21) | -0.95 (-0.29) |
| Shelters | 0.06 (0.28) | -0.32 (-0.57) | 0.38***(9.80) | 0.25 (0.31) | -0.50 (-1.17) | -0.47 (-0.75) |
| Fire and police services | -0.11*(-1.67) | 0.51 (0.97) | -0.004 (-0.08) | 0.15 (0.23) | 0.16 (0.54) | 0.79 (1.23) |
| Medical services | -0.02 (-0.79) | -0.38***(-5.64) | -0.04*(-1.85) | -0.50***(-5.60) | -0.32*(-1.65) | 0.01 (0.05) |
| Access to resources | 2.79***(2.88) | -11.91***(-2.72) | — | -30.97***(-3.90) | — | -16.57***(-3.16) |
| Adaptation appraisal | 0.82 (1.20) | -4.29***(-2.71) | 1.25***(3.12) | -7.76***(-3.32). | -1.41*(-1.66) | -1.58 (-0.74) |
| Alpha | 1.16***(2.57) | — | — | — | — | — |
| $\chi^2$ | 36.30*** | — | 5571.26*** | — | 359.82*** | — |
| Log-likelihood function | -219.62 | -1692.15 | -196.38 | -1093.23 | -44.10 | -566.21 |
| LM test | — | 325.37*** | — | 223.43*** | — | 174.69*** |
| Sample size | 598 | | | | | |

[a]: $Z$-test value in parentheses; *: significant at $p < 0.1$;**: significant at $p < 0.05$; ***: significant at $p < 0.01$