# Peer review of "Linking local vulnerability assessments to climatic hazard losses for river basin management"

_Natural Hazards and Earth System Sciences, 2016_

## Referee Comment (RC1) · Anonymous Referee #1 · 4 Jul 2016

**To the authors and editors:**

This paper deals with the losses and damages of a typhoon and relates these to a composite vulnerability index. The underlying datasets and the statistical work is of relevance for the scientific community. However, there are significant lacks in

- The conceptual frame for the work
- A critical view of the approach followed and of the results achieved.

In addition, the text has to be significantly improved regarding the English language. I have given a number of proposed correctinos in the first half of the text (See below).

I would encourage the authors to review their paper thoroughly and particularly regarding the various concepts of vulnerability. I would also like to ask them to take the constraints of their methodologies into consideration when discussing their results.

**General comments to the paper:**

The concept of vulnerability and the implication that the conceptual approach has on the study is not clear:

- At the end of chapter 2.1 the authors state that it is necessary to integrate the vulnerability concepts of the disaster community and of the IPCC. The proposed formular (1) however does reflect only the IPCC concept. If the authors start to discuss these conceptual issues they need to be much more sharpened in their explanation of the differences of the various approaches and why and how they would like to integrate approaches
- The two approaches for investigating the relationships between vulnerability and disaster losses in chap 2.2 are not described clearly enough.
- The methodology for the selection of indicators is not transparent. There is a lack of clarity in the concept reflected in the description of the indicators in chapters. 3.1.X. For example, coping is mentioned as part of both sensitivity and adaptive capacity.
- A critical reflection on the selection of a limited number of indicators is missing
- A discussion of the problems when using statistical methods when only limited damage and loss data is available is entirely missing.
- A description of the typhoon event itself is missing
- It is not clear for which spatial extend the regression analysis has been carried out. For example, what was the spatial resolution of rainfall data? How did the authors deal with the fact that the data is available in different formats (point, raster etc).
- The MCDA has not been described in detail, what is it exactly and which role does it play?
- The discussion needs to consider the problem to look at hazard and vulnerability factors separately. The authors state that "villages with higher elevations, in upper streams and more proximity to rivers tended to suffer more disaster casualties and losses due to their higher exposure to typhoon 3 impacts". Unclear remains wht the difference is

between exposure and typhoon impacts (are impacts = damage?). Then they conclude, "However, constraints associated with local government adaptation efforts in the river basins reflect a range of challenges in relation to how the integrated RBM adaptation efforts have structured. The efforts to facilitate adaptation should largely target the mitigation of vulnerability and risk." – these types of conclusions need to be explained further.

Specific comments in the text:

**Linking local vulnerability assessments to climatic 1 hazard losses for the river basin management 2**

3 4 5 6 7 8 By 9 Hung-Chih Hung1(Corresponding author) 10 1 Professor, Department of Real Estate and Built Environment, National Taipei 11 University, New Taipei City,23741,Taiwan12 Address: 151, University Road, San-ShiaDistrict, New Taipei City, 23741, Taiwan13 Phone: +886-2-8674-1111 ext.6743314 Fax: +886-2-8671-530815 Email: hung@mail.ntpu.edu.tw16 Yi-Chung Liu217 2National Science and Technology Center for Disaster Reduction, New Taipei City, 18 23143, Taiwan19 Email: ycl@ncdr.nat.gov.tw20 Sung-Ying Chien321 3 National Science and Technology Center for Disaster Reduction, New Taipei City, 22 23143, Taiwan23 Email:csy0904@ncdr.nat.gov.tw24 25 26 27 1

Abstract. To prepare for and confront the potential impacts of climate change and related 1 hazards, many countries have implemented programs of integrated river basin 2 management. This has led to an imperative challenge for local authorities to improve 3 the understanding of how the vulnerability factors link to climatic disaster losses. This 4 article aims to examine whether high vulnerable areas experience significantly more 5 damage caused by weather extreme events at the river basin levels, and explain what 6 vulnerability and hazard impact factors determine the disaster losses. Using three river 7 basins in southern Taiwan attacked by Typhoon Morakot in 2009 as case studies, we 8 proposed a novel methodology that combined a geographical information system (GIS) 9 technique with a multicriteria decision analysis (MCDA) to evaluate and map 10 composite vulnerability to climatic hazards across river basins. Then, the linkages 11 between hazard impacts, vulnerability factors and disaster losses were tested by using 12 a disaster damage model (DDM). The results of the vulnerability assessments 13 indicated that the vast majority of the most vulnerable areas is situated in the regions 14 of middle, upper reaches and some coastlines of the river basins. Using the DDM, it 15 shows that the typhoon losses and casualties are significantly influenced by local 16 vulnerability contexts and hazard impact factors. Finally, we suggest the implications 17 of adaptation policy lines for minimizing vulnerability and risk, as well as for 18 integrated river basin governance. 19

**1** Introduction 20**

Major portions of Asia have an increasing exposure and vulnerability to climate 21 change and weather extremes due to rapid urbanization and overdevelopment in 22 hazard-prone areas (IPCC, 2014). For example, in August 2009, a devastating 23 typhoon (Morakot) hit three major river basins in southern Taiwan. Meanwhile, 24 approximately 700 people were killed and total economic losses were estimated to 25 2

Comment [S1]: More than what? Needs to be reformulated Comment [S2]: Language check

Comment [S3]: propose ?

Comment [S4]: ??, please clarify

Comment [S5]: Language check

**Comment [S6]:** This is not an appropriate example for the statement in the first sentence since this is one event and does not tell as anything about recent trends

have been US\$ 0.6 billion (Liu et al., 2014). Thus, it becomes increasingly important 1 for water resource managers to implement programs of integrated river basin 2 management (RBM) that can cope with and reduce potential impacts of climate 3 change and climate-related (climatic) disaster risk (Hung et al., 2013). 4

Integrated water resource management is a process to promote the coordinated 5 development and management of water, land uses and related resources (GWP 2000). 6 This indicates that the integrated RBM should adopt the river basin as a management 7 unit, employing a comprehensive perspective to connect water resource management, 8 agricultural irrigation with land use planning for building more resilient river basin 9 contexts (Penning-Rowsell et al., 2006). Especially, vulnerability assessment plays a 10 vital role in scrutinizing the biophysical, socioeconomic conditions and their 11 distributions over river basins. This process of assessment also helps decision makers 12 integrate various local connections into planning and policy lines for disaster damage 13 and risk mitigation within the context of whole river basins (Hooijer et al., 2004; 14 Hung and Chen, 2013; You and Zhang, 2015). 15

Existing vulnerability analyses have focused more on assessing, mapping and 16 distinguishing the variability of the vulnerability distribution among regions (Adger, 17 2006; Hung and Chen, 2013; Ahumada-Cervantes et al., 2015). However, climatic 18 disaster loss and risk accumulation result from the interlinking between hazard 19 impacts, exposure and vulnerability components (UNISDR, 2012; Hung et al., 2013). 20 The majority of extant studies inferred disaster losses and risks use computer-aided 21 simulation, scenarioanalyses and

multicriteriadecisionanalysis(MCDA) (Tate et al., 22 2010; Ni et al., 2010; Hung et al., 2013; De Bruijn et al., 2014). Theirfindings are 23 valuablein characterizingdisaster risk, impacts and their distributions that enable 24 decision makers to create risk maps and communicate the highriskareas to 25 3

Comment [S7]: and ?

**Comment [S8]:** language – reformulate or give relation: more than?

Comment [S9]: Why 'however'?

**Comment [S10]:** Language, pls reformulate

Comment [S11]: Existant ?

Comment [S12]: Language check

stakeholders. Nonetheless, few studies have systematically examined how the 1 vulnerability and hazard impactfactors link totheir potential influence on disaster2 losses. This would

compromiscompromise ethe application of existing vulnerability and exposure 3 studies tothedisaster risk assessmentand integrated RBM. 4

This article aims to examine whether geographic-localities characterized by high 5 vulnerability experience significantly more damage owing to onset weather extreme 6 events at the river basin level, and to explain what vulnerability, hazard and exposure 7 factors influence these damages or

losses. Using three river basins in southern Taiwan 8 hit by Typhoon Morakot as case study areas, we propose a novel methodology based 9 on existing disaster impact theory, which then combined an MCDA, GIS 10 (geographical information system)-based statistics with multivariate analysis to assess 11 climatic hazard vulnerability (especially typhoon and flood). Moreover, we examine 12 the connection between vulnerability, hazard impact factors and disaster losses using 13 a disaster damage model (DDM). The methodology may also be applicable to other 14 river basins. Finally, we discuss the extension of our findings in providing policy 15 directions for building adaptive capacity and for integrated RBM. 16

**2 Vulnerability and disaster impacts17**

**2.1 Vulnerability assessment 18**

Vulnerability assessments haves been broadly applied to various research communities 19 with respect to climate change adaptation and disaster risk management, although not 20 agreeing on a common view about the concept of vulnerability. In the disaster impact 21 research, vulnerability is widely described as the degreesof susceptibility of these 22 assets to sufferdamage and loss (UNISDR, 2013). Furthermore, IPCC (2014) has 23 conceptualized vulnerability as a variety of concepts and elements that encompass 24 4

Comment [S13]: Add references pls

Comment [S14]: Language check

sensitivity or susceptibility to harm and lack of capacity to cope and adapt. 1 Watersheds' contexts consist of biophysical, socioeconomic, industrial and land use 2 elements. Thus, from the perspective of integrated RBM, vulnerability assessment 3 should facilitate decision-makers to engage in the integrated analyses of interaction 4 between the components of vulnerability and the properties of a specific watershed 5 context (O'Brien et al., 2007; Engle and Lemos, 2010; Hung and Chen, 2013). To 6 target support integrated RBM, it should integrate IPCC's (2014) with UNISDR's 7 (2013) concept to build more transdisciplinary and comprehensive vulnerability 8 assessment framework. Therefore, the vulnerability can be generally described as a 9 function of exposure, sensitivity and adaptive capacity: 10 Vulnerability = f (exposure, sensitivity, adaptive capacity) (1) 11

**2.2 Vulnerability and disaster losses 12**

Existing approaches on the investigation of the relationships between vulnerability 13 and disaster losses can be divided into two major types. Thefirst type of approach 14 interpretsdisaster damageor risk as a function of vulnerability, and frequentlyusing 15 the catastrophic, PSR (pressure-state-response), PAR (pressure-and-release) theories 16 or MCDAcoupledwith computer-aid simulationand GIS-based analysesto predict 17 disaster losses(Ermoliev et al., 2000; Wisner et al., 2004; Tate et al., 2010; Scheuer et 18 al., 2011; De Bruijn et al., 2014). Therefore, disaster risk or potential losses can be 19 directly projected by vulnerability assessment (Cutter et al., 2003; Hung and Chen, 20 2013). [This type of researchuses a 'top-down' approach that can bringthe disaster 21 information related to predicted distributions of disaster impacts and riskto the fore, 22 although there are uncertainties and ambiguities in the processes of projection 23

The second type of approach focuses on the 'bottom-up'and data-based analysis, 24 5

**Comment [S15]:** Language check, unclear

**Comment [S16]:** Needs clarification and more sharpened explanation. Fore example ok that Wisner et al. interprete losses as a function of vulnerability. But first, this vulnerability is differently understood then the vulnerability of the authors' formular (1) and second, Wisner would for sure not predict disaster losses with computer / GIS techniques... Also lang issues – why is it

'catastrophic?'.

**Comment [S17]:** If the authors' have found this strong statement in the cited references, please give the page number (or best a word-forword citation. If this statement is supported by the authors' finding it needs more explanation

**Comment [S18]:** Unclear, lang check needed, what are the main reasons for uncertainty and ambiguity ?

which often uses historical or surveyed data to characterize the disaster damage1 (Zahranet al., 2008; Bhattaraiet al., 2015). The findings not only can help decision 2 makers identify disaster loss distributions, but also examine their determinants 3 (Downton and Pielke, 2005). Thisapproachconcentrates more on mapping the 4 disaster damage distribution at the national or regional levels. It also majorly 5 combines expert judgment with mono-dimensionalevaluation to inspect the 6 influentialfactors of disaster damage(Mokrech et al., 2012; Hung and Chen, 2013). 7 However, little attention had been paid to linking multi-dimensional vulnerability 8 assessment with empirically-based disaster loss evaluation inthe river basin contexts. 9

Using theoretical-based simulation and MCDA, the first type approach seeks to 10 systematically identify disaster loss and scrutinizeits components, as well as to 11 project various

disasterimpactsresulting from different hypotheticalevents. By 12 contrast, thesecond type approach enablesdecisionmakers a conjoint treatment of 13 quantitativedisaster loss data and qualitativehuman judgment. Nonetheless, these two 14 types of approachesall considering disaster losses are inherent and dynamic due to 15 ongoing interaction of climatic hazard impacts with the biophysical and 16 socioeconomic dimensions of vulnerability in awatershed system (O'Brien et al., 17 2007; Maru et al., 2014).18

Increasing the understanding of the formation of climatic disaster risk highlights 19 the importance of connecting a forementioned two types of approaches and their 20 relative

magnitudes(Mokrechet al., 2012; Visseret al., 2014). Particularly, 21 incorporating the first type into the second type approach allows us to create 22 frameworks of disaster risk analysis that could assist in expanding the range of 23 vulnerability assessments and in sequencing them to generate robust resilience and 24 adaptation pathways (Hung et al., 2016).25 6

**Comment [S19]:** How? By looking at loss and damage data? Or by using expert opinion as described below? Than the sentence below should come up. Please clarify

**Comment [S20]:** More than?? See comments above

**Comment [S21]:** What is this? It has not been discussed in the chapter where the authors debate the vulnerability concept

**Comment [S22]:** What is this? Pls explain

**Comment [S23]:** You mean *those* two types?

**3 Methods and data1**

To characterize the disaster loss distributions and their linkages with various 2 vulnerability components, we incorporated an MCDA and GIS-based statistic analysis 3 into a data-based multivariate analysis. First, the composite vulnerability framework 4 was constructed to summarize a review of the literature and combined with an MCDA 5 to assess climatic hazard vulnerability at the river basin level. Second, based on PSR 6 framework, the relationship between disaster loss distributions, impacts and 7 vulnerability factors was tested and compared using numerous regression models. 8 Finally, we discussed the findings and provided implications for better adaptation 9 policy lines. 10

**3.1 Indicators of vulnerability framework and hypotheses 11**

The assessment framework created here was based on the IPCC's (2014), UNISDR's 12 (2013) concepts of vulnerability and literature review. This framework allows us to 13 take advantage of the contributions of existing knowledge, as well as obtain synergies 14 and complexities of watershed contexts as discussed in detail in Hung and Chen 15 (2013). We identified the indicators involved in the framework consisting of three 16 dimensions: exposure, sensitivity and adaptive capacity. In terms of assessing the 17 integrated vulnerability, we mainly adopted the framework of vulnerability indicators 18 promulgated by Hung and Chen (2013), which was appropriate and widely applied to 19 the river basin conditions in Taiwan. Then, an assessment of composite vulnerability 20 was conducted across the case study areas at the village scale, which is the basic unit 21 of local administration in Taiwan. 22

3.1.1 Exposure indicators 23

Exposure refers to the biophysical factors and the extent to which properties of 24 7

**Comment [S24]:** Check language, please reformulate**

**Comment [S25]:** What does this mean? That you selected your indicators based on literature review? Please be clearer

**Comment [S26]:** If the PSR plays such a crucial role it needs to be shown in a figure. Based on this figure you should then clarify between which components you try to describe relations by regression models

**Comment [S27]:** This is the first time you mention indicators, what are they indicating? Why did you select this method?

**Comment [S28]:** Ok but needs a bit more explanation, you must be able to understand this paper without having read Hung and Chen (2013), vulnerable system are in contact with hazards(Hung et al., 2016). To reflect the 1 degrees of exposure, *averaged annual rainfall*, and *potential debris flow torrents* were 2 used. The expectation is that either higher rainfall or debris flow torrents would 3 increase vulnerability and thus enhance the likely disaster losses (Wisner et al., 2004). 4

**3.1.2 Sensitivity indicators 5**

Sensitivity is one of the most broadly used attributes to describe the vulnerability in 6 climate change and disaster risk management (Cutter et al., 2003; O'Brien et al., 7 2014). The sensitivity indicators are mostly composed of inherent biophysical and 8 societal contexts, and the societal context can be further classified into socioeconomic 9 and land use sensitivity (Hung and Chen, 2013). 10

The hypothesized links between biophysical context and disaster losses are 11 captured in examining the influence of *proximity to rivers* and *elevation* indicators on 12 disaster losses. The areas where are more proximity to rivers and/or at higher 13 elevations are both the more vulnerable and sensitive to disaster damage (Ni et al., 14 2010). The socioeconomic indicators include *populations, social dependence, income,* 15 *employment* and *production values* of industries and services. These indicators are apt 16 to reflect the extent of areas' contextual vulnerability and fragility in a watershed. 17 Thus, increasing income, employment and/or production values by communities is 18 expected to enhance coping strategies, thereby decreasing vulnerability and potential 19 disaster losses (Zahran et al., 2008). Contrarily, populations and social dependence 20 have expected a positive relation to disaster damage (Hung et al., 2016). 21

In the aspect of land-use, the indicators comprise *urban developments, agricultural* 22 *uses, envornmental sensitive areas* and *road infrastructures*. Generally, while 23 preserving more sensitive areas could decrease vulnerability and disaster losses, the 24 larger scales of either urban developments, agricultural uses or road infrastructures 25 8

**Comment [S29]:** But Wisner would not say that rainfall is part of vulnerability since he is following a different concept. Ok if you include in your V concept the parameter rainfall (or other bio-physical paramters) but you cannot really give Wisner as a reference. It all goes back to the point that the description of vulnerability in 2.1 needs to be improved.

**Comment [S30]: Language check**

**Comment [S31]:** In your concept you say that sensitivity is part ov vulnerability. Therefore this does not make sense here

**Comment [S32]:** Fragility has not mentioned before – wehre dos this fit in your picture of vulnerability?

would encourage denser land, agricultural developments and more tourist activities, 1 and that could lead to higher vulnerability and expected disaster losses (Cutter et al., 2 2003; Mehaffey et al., 2008). 3

**3.1.3 Adaptive capacity indicators 4**

Using adaptive capacity indicators to measure the ability of communities to adjust to 5 potential damage, to take advantage of opportunities, or to respond to disaster 6 consequences (IPCC, 2014), the indicators include shelters, medical, fire and police 7 services. These indicators present an area's abilities of coping, evacuation and 8 emergency responses. Therefore, improving these facilities could reduce vulnerability 9 and likely disaster damage. We also involved behavior and heuristic factors that 10 consisted of residents' risk perceptions, their ability to access to resources and to 11 successfully adapt to hazards (self-efficacy). The hypothetical relationships between 12 these factors and disaster damage are negative (Eakin et al., 2010). Finally, those 13 indicators considered to assess vulnerability are demonstrated in Table 1, along with 14 their descriptions, data sources, and expected directions of relations to disaster losses. 15 3.2 Composite vulnerability index 16

To assess the vulnerability for each village, the composite vulnerability index (CVI) 17 was estimated. However, the surveyed values of various indicators contain different 18 scales and units. We applied a min-max scaling to directly normalize all the data into 19 a uniform [0, 1] scale with ratio properties. Then, the normalized values were used to 20 computeCVIsby:21  $CVI_i = (2)22 \square \square mj ij j x w 1$ .

Where CVI/represents the composite vulnerability index for village i; xij denotes the 23 9

Comment [S33]: This should not come under one component only

Comment [S34]: This needs an introduction, or just formulated differently since the explanation follows below

normalized value for indicator *j*, and *wj* is the weight. With above hypotheses, if  $x_{ij} > 1$  0, it indicates higher levels of overall vulnerability; if  $x_{ij} < 0$ , decreasing or lessening 2 the overall vulnerability. Furthermore, equal-weight was assigned to each indicator in 3 order to build an equivalent basis for comparing the attributes of vulnerability and 4 disaster losses among the river basins. 5

**3.3 Linking vulnerability factors and climatic disaster losses 6**

This study focuses on single-scenario disaster eventfor comparativelystatic 7 modelling ofdamages and losses at differentpointsover river basins. This approach 8 allows us to controlthe disaster scenario, so thatany variationin lossescan be 9 directly resulted from changesin hazard impactsand vulnerability factors(Hung et al., 10 2013). Therefore, the disaster damage model can bewritten asthe following function: 11

Disaster loss = f(hazard, vulnerability)

Equation (3) implies that the interaction of hazard impacts and vulnerability 13 generates disaster losses. Therefore, the extent of disaster damage and/or losses will 14 vary with vulnerability contexts, while climatic hazard (i.e., typhoon event) impacts 15 are deemed as outerdependant factors. To more specifically identify the relationship 16 between disaster losses and vulnerability factors, several regression models were used 17 in the case studies. 18

(3)12

**3.4 Case study areas and data 19**

This article explores three very different river basins, choosing for representing the 20 areas with various degrees of development and contexts in southern Taiwan (Fig. 2), 21 but all having heavily struck by the Typhoon Morakot in 2009. The case study areas 22 include three major river basins: Gaoping, Tsengwen, and Taimali Rivers. According 23 to the 2015 census, these three river basins encompass598 villages, around 1.26 24 10

**Comment [S35]:** Ok, now we have the disaster community concept of risk where hazard and vulnerability are considered separately. But in the concept used for V earlier the rainfall has been included. This does not fit to the formula 3 here

Comment [S36]: Lang check

million populations and cover an area of approximately 7,885km2. Highly diversified1 topography distributes over thethreewatersheds. The altitude of this region ranges 2 from the coastal lowlands along the western shoreline to over 3,000 meters in the 3 eastern high-mountain areas. Uncontrolled urban sprawl and environmental 4 destructionare interwoven by growing threats from climate change and weather 5 extremes that lead to the riskiestregions in Taiwan(Liu et al., 2013). 6

In modelling the linkagesbetween vulnerability factors, disasterimpacts and losses, 7 the data were collectedfrom multiple sources. The disaster loss database regarding 8 Typhoon Morakot had been systematically built by the Department of Science and 9 Technology, Taiwan. Thisdatabase included the surveyed numbers of casualties, 10 property and agricultural losses, the distributions of inundation and landslide, and 11 damaged public facilities. The data on vulnerability factors were obtained through 12 combing official statistic censuses and random sampling face-to-face questionnaire 13 survey to residents (shown in Table 1 in detail). 14

**4 Results and discussions15**

**4.1 Composite vulnerability assessments 16**

Using the CVIs estimated by equation (2), Fig. 1 shows the distributions of estimated 17 index values superimposed on the geopolitical boundaries of villages throughout the 18 three river basins. The CVI estimates were divided into five levels (at 20% intervals). 19 The villages with the estimated index values within the 80-100th percentiles can be 20 defined as the most vulnerable, and within the 1st-20th percentiles as the least 21 vulnerable. 22

In Fig. 1, it shows that there are highly heterogeneous in the spatial distributions of 23 estimated \_\_\_\_\_\_ compositevulnerability across the study areas. In the Tsengwen River basin, 24 11

**Comment [S37]:** Rather administrative

Comment [S38]: Language check

the most vulnerable areas concentrated in the middle reaches and some coastlines. 1 Moreover, most the middle and upper reaches of the Gaoping and Taimali River basin 2 (especially northern shore) were distributed by the most vulnerable villages, while 3 most of the lower reaches were spread with the least vulnerable ones. These spatial 4 distribution patterns highly conform to historical experience with which numeral 5 typhoons had hit these areas in past years, and resulted in serious casualties, property 6 and crop losses. 7

The results corroborate similar findings from related studies (Hung and Chen, 2013; 8 Liu et al., 2013), asserting that a significantly specially-defined clusters of highly 9 vulnerable areas are mostly situated in midstream and upstream reaches. This leads to 10 a challenge for watershed managers in understanding of why these areas are 11 particularly vulnerable and how they link to disaster losses, as well as what the 12 implications of this might be for land-use planners to reduce risk. 13

**4.2 The distributions of disaster losses due to Typhoon Morakot 14**

The inundation areas due to Typhoon Morakot concentrated in the convergent regions 15 of Kaoping River and its tributaries, while the major landslide and debris flow torrents 16 occurred in the middle and upper reaches. This would affect the distributions of 17 property, public facility and agricultural damage (Fig. 3). Using *t* test for correlation 18 analysis, it showed that the location of agricultural damage significantly corresponded 19 to where the landslides (Spearman

= 0.18, p < 0.01; Pearson r = 0.43, p < 0.01) and 20 damaged bridges occurred. The pattern of casualties also highly correlated with the 21 numbers of landslides (Spearman = 0.22, p < 0.01, Pearson r = 0.23, p < 0.01) and 22 damaged bridges (Spearman = 0.40, p < 0.01, Pearson r = 0.42, p < 0.01). 23

In the Tsengwen river basin, the impacts of flooding and landslides caused more 24 serious damage to the watersheds than debris flow torrents. This would lead to that 25 12

Comment [S39]: Spatially?

**Comment [S40]:** A short description of the typhoon would be helful

both casualty counts and agricultural losses significantly associated with patterns of 1 landslides (casualties: Spearman = 0.17, p < 0.05, Pearson r= 0.53, p < 0.01; 2 agriculture: Pearson r= 0.56, p < 0.01) and damaged bridges (casualties: Spearman = 3 0.27, p < 0.01, Pearson r= 0.55, p < 0.01; agriculture: Pearson r= 0.40, p < 0.01). 4 Agricultural and property losses in the Taimali watershed were mostly agglomerated 5 along the road systems. It indicates a noteworthy relationship between road 6 infrastructures, land developments and disaster losses that needs further investigation. 7

**4.3 The determinants of disaster losses 8**

The regression analyses for examining the determinants of typhoon losses include 9 casualties, property and agricultural losses. The choice of regression models was 10 based on the distribution types of disaster loss data. The distribution of disaster 11 casualties is non-normal. Zero counts significantly skew the distribution leftward– 12 93% of Typhoon Morakot caused no recorded injuries or fatalities. The total 13 casualties are 684, the arithmetic mean is 1.01 and the standard deviation is 18.76 – 14 dispersion is 18.6 times greater than the average. The casualties are a non-negative 15 integer exhibiting significant over-dispersion with a disproportionate number of zero 16 counts, we thus investigated the data using a ZINB (zero-inflated negative binomial) 17 or ZIP (zero-inflated Posisson) regression model, which allows us to estimate the net 18 effects of independent vulnerability factors on casualties (Cameron and Trivedi, 1998; 19 Zahran et al., 2008). To more comprehensively scrutinize the influence of disaster 20 losses, the integrated typhoon loss index (ITLI) was estimated to serve as proxies for 21 combined losses of typhoon: 22

**ITLIi = Agiculturei + Propertyi + Casualtyi. (4) 23**

Where *Agiculturei* and *Propertyi* are agricultural and property losses for village *i*, 24 respectively; *Casualtyi* is casualty counts. A Lagrange multiplier (LM) test points to 25 13

evidence of which the ITLI is a non-negative rational number significantly spreading 1 in a certain range. Thus, we applied a Tobit (Censored) regression model to examine 2 the affecting factors of ITLI. 3

Table 2 reports the results of ZINB and ZIP regression analyses for typhoon 4 casualties, as well as Tobit models for ITLIs. Six separate models are estimated, with 5 predictors both for each watershed (excluding Taimali River due to little sample size) 6 and for all three river basins. To screen variables for multicollinearity, we used 7 zero-order correlation and Variance Inflation Factor tests in Ordinary Least Squares 8 regression. It showed that the *risk perceptions* and *access to resources* have 9 significantly higher multicollinearity with other variables. These two variables were 10 thus eliminated in some regression analyses.11

In all regression models, results indicate that most hazard impact factors play an 12 important role in determining typhoon casualties and losses. As expected, the 13 landslides, damaged bridges, agricultural losses, property losses and flooding areas 14 are positively associated with typhoon losses, although agricultural losses are 15 negatively related to casualties in Gaoping watershed. These findings correspond with 16 the PSR framework that could consider the hazards as pressures and their impacts 17 would change the quality of the environment. The higher the hazard impacts, the 18 higher the odds of casualty and disaster loss (OECD, 1993; Wisner et al., 2014). 19

Regarding the biophysical exposure indicators, averaged rainfall was a major 20 positive contributor to the casualty counts in both Gaoping and Tsengwen watersheds, 21 while it was a negative predictor of disaster losses. In Gaoping River, the high 22 casualties occurred in the areas with higher levels of rainfall and elevations rather 23 than in debris flow torrents distributed areas. The areas within 0-200m to rivers 24 significantly increased the numbers of casualty over three river basins, and enhanced 25 14

typhoon losses in both Gaoping and Tsengwen watersheds. Most of these results are 1 consistent with our expectation and earlier studies on the linkage between biophysical 2 factors and disaster losses (OECD, 2012; Hung et al., 2016). It implies that the areas 3 with higher risk are mostly located in the regions with higher elevations and more 4 proximity to the rivers over the watersheds. 5

In the compilation of socioeconomic factors, population density was a strong 6 predictor of casualty counts and disaster losses, and was negatively related to casualty 7 counts, while its relation to disaster losses was positive (excluding Gaoping River). 8 Findings reflected that the patterns of disaster damage would depend on the types of 9 hazard impacts. The upstream areas were frequently distributed with the low density 10 population, but more landslides occurred, and that would cause higher casualties. 11 Generally, the inundation was mostly assembled in downstream areas, which would 12 lead to more overall losses than casualties. 13 The lower income areas were likely generating more casualties. Furthermore, as 14 one enhances

the production values of industries and services in an area, the increase 15 of the capacity of predisaster preparedness and emergency responses that can 16 decrease disaster loss and risk. Except for a significantly positive relationship between 17 employment rates and casualty numbers, and between social dependence counts and 18 disaster losses in Gaoping watershed, the other socioeconomic factors had a weak 19 relation to casualty and loss distributions. These results do not fully conform to 20 existing studies that highlight the relationship between social vulnerability factors and 21 disaster losses (or risk) is functional (Zahran et al., 2008; Hung and Chen 2013). 22 Rather, their relations have remained complex and difficult to model, depending on 23 multiple influences of local contexts and disaster impacts involved in each watershed 24 (UNISDR, 2012). 25 15 In all regression models, mounting urban or agricultural developments significantly 1 increased casualties and typhoon losses, although increasing agricultural uses strongly 2 decrease casualty distributions in Gaoping watershed. These results also reflected in 3 that more sensitive areas reserved could reduce the occurrence of casualties and losses 4 (excluding Tsengwen watershed). In addition, provision of road or transportation 5 infrastructures would be helpful in the evacuation and disaster relief, and lead to 6 fewer casualty counts after typhoon hitting. As most extant studies emphasized 7 (Mehaffey et al., 2008; Hung et al., 2016), our case study shows the evidence that 8 higher levels of urbanization and farming reclamation would increase hazard 9 vulnerability and further result in higher damage. 10

Concerning the adaptive capacity variables, they also played a critical role in 11 predicting disaster damage. Especially, increasing medical services, access to 12 resources and self-efficacy significantly attenuated the disaster losses, as well as 13 strongly decreased casualty counts in both Gaoping and Tsengwen river basins. These 14 results confirm the earlier findings that emphasizing the improvement of adaptive 15 capacity could effectively reduce disaster damage and risk (Eakin et al., 2010; Hung 16 and Chen, 2013). However, one noteworthy exception is that the areas with higher 17 ability to access to resources had been distributed with more typhoon casualties across 18 the three river basins. One possible explanation is that most these areas are 19 particularly vulnerable and frequently received large amounts of external aids in the 20 aftermath of a disaster hitting. However, these exterior aids might be valuable in 21 temporary disaster relief rather than improving long-term vulnerability. 22

**4.4 Policy implications 23**

This research presents a systematic starting point to investigate a novel topic on the 24 relationship between vulnerability attributes, hazard impacts and losses. Through the 25 16

Comment [S41]: ??

composite vulnerability assessments and regression analyses, it shows that the 1 villages with higher elevations, in upper streams and more proximity to rivers tended 2 to suffer more disaster casualties and losses due to their higher exposure to typhoon 3 impacts. However, constraints associated with local government adaptation efforts in 4 the river basins reflect a range of challenges in relation to how the integrated RBM 5 adaptation efforts have structured. The efforts to facilitate adaptation should largely 6 target the mitigation of vulnerability and risk. Especially, combining resilient types of 7 infrastructure, warning system with risk communication to improve the emergency 8 system is essential for predisaster hazard-mitigation planning that helps reduce risk 9 and save the lives (Hung and Chen, 2013; Hung et al., 2013). 10 In the long-term policy lines for integrated RBM that the land use planning coupled 11 with regulation, relocation and building codes can help restrain urban and agricultural 12 developments encroaching onto hazard-prone areas(Neuvel and van den Brink, 2009). 13 As the vulnerability distributions and their linkages to disaster losses presented in this 14 study, it enables the policy makers to generate hazard risk maps that provide a useful 15 initial step to identify and communicate the riskiest areas to stakeholders. In the upper 16 streams, land use management can be further integrated into river basin governance in 17 order to keep the environmental sensitive areas from excessive urban sprawl, 18 agriculture and tourism activities, as well as to appraise adaptation options for the 19 most vulnerable areas. Besides structural engineering projects, the downstream areas 20 need to incorporate wetland preservation, flood insurance, warning system and related 21 risk-sharing arrangements into the existing RBM framework forminimizingrisk. 22

**5 Conclusions23**

Growing climate change and weather extreme impacts pose impending challenges and 24 high uncertainties for the RBM. Therefore, the understanding of the interlinks 25 17

between disaster impacts, vulnerability factors and losses is critical for disaster risk 1 and river basin governance within the options of which adaptive strategies take place. 2

This article proposes a novel approach that stems from the combination of previous 3 studies on vulnerability assessments and disaster impacts to unpack and characterize 4 the vulnerability over river basins, and to examine its influence on typhoon losses. A 5 composite vulnerability assessment framework was constructed in hybrid with an 6 MCDA to create vulnerability maps that can be valuable to inform policy-making and 7 communicate the core areas in which adaptive measures are most needed to reduce 8 vulnerability and risk. Applying various regression models to examine the key 9 vulnerability and hazard impact factors that determined the casualties and losses 10 caused by Typhoon Morakot, as well as compare the typhoon losses between river 11 basins due to the variability in local contexts. 12

The findings indicate that both the hazard impacts and vulnerability factors can 13 strongly vary spatial distribution patterns of disaster losses. Especially, local 14 biophysical, socioeconomic and land use attributes are key predictors to disaster 15 losses. Local agencies should make some tradeoffs between building adaptive 16 capacity and reducing vulnerability. However, the disaster event considered in this 17 study is limited. Further case studies across other river basins can provide more 18 insights into how crucial the tradeoffs may be to reduce risk. Moreover, the 19 robustness and application of our modelling need to be examined by comparing the 20 operationalized loss surveys of additional cases in the aftermath of other disaster 21 events. This is able to offer the integrated RBM with some useful policy and land use 22 planning indications in building more resilient river basins. 23

**References24**

Adger, W. N.: Vulnerability, Glob Environ Chang, 16, 268-281, doi: 25 18

10.1016/j.gloenvcha.2006.02.006, 2006. 1

[revised manuscript text omitted]

(2) Taimali River basin(3) Tsengwen River basin 22

(1) Distributions of damaged bridges (2) Distributions of casualties Fig. 2 Distributions of the losses due to Typhoon Morakot over three river basins 23

| Table 1Hazard
impacts,
vulnerability
indicators
(variables)
andexpected
sign
toclimaticdisast
er
lossesCategory | Indicator                                           | Descri                             | ption                          | Data sou               | irce              | Mean (S.D.)  | Sign |
|--------------------------------------------------------------------------------------------------------------------------------------------|-----------------------------------------------------|------------------------------------|--------------------------------|------------------------|-------------------|--------------|------|
| Hazard                                                                                                                                     | Casualties                                          | Numbe                              | r of                           | NCDR a ,    | Taiwan            | 1.01 (18.76) | +    |
| impacts                                                                                                                                    |                                                     | casualti                           | ies(people                     |                        |                   |              |      |
| Landslides                                                                                                                                 | Areas of lands
(km2)                             | lides                              | NCDR, Tai                      | wan                    | 0.48 (2.          | 22)          | +    |
| Damaged bridges                                                                                                                            | Number of dat
bridges                            | maged                              | NCDR, Tai                      | wan                    | 0.24 (0.          | 90)          | +    |
| Agricultural losses                                                                                                                        | Amount of
agricultural los
(1000 NT\$)        | sses                               | NCDR, Tai                      | wan                    | 14.01 (3          | 39.34)       | +    |
| Property losses                                                                                                                            | Number of dat
dwelling                           | maged                              | NCDR, Tai                      | wan                    | 48.68 (1          | 26.8)        | +    |
| Flooding areas                                                                                                                             | Areas of inunc
(km2)                             | lation                             | NCDR, Tai                      | wan                    | 0.32 (0.          | 94)          | +    |
| Exposure                                                                                                                                   | Rainfall                                            | Averag
rainfall                 | edannual
(mm)               | Central W
Bureau, 7 | Veather
Faiwan | 1932(364)    | +    |
| Debris flow torrents                                                                                                                       | s Number of pot
debris flow to
and landslides | tential
rrents                  | Council of
Agriculture      | , Taiwan               | 0.41(1.0          | )7)          | +    |
| Sensitivity Bi                                                                                                                             | ophysical Proz
ntext rive                        | ximity to
rs                    | Areas w
0m-2001
torivers | ithin 1
n (km2)     | Measured t
GIS | oy 0.18 (0.2 | 1) + |
| Elevation                                                                                                                                  | Averagedeleva
(m)                                | ation                              | Ministry of
Interior, Tai   | the
wan             | 169.7(3           | 55.3)        | +    |
| Socioeconomic sensitivity                                                                                                                  | Populations                                         | Populat
density
(popula
) | tion
ntions/km2             | Ministry
Interior,  | of the
Taiwan  | 2.74 (5.60)  | +    |
| Social dependence                                                                                                                          | Ratioof people
age 65 and une
6, and females  | e over
der age
(%)           | Ministry of
Interior, Tai   | the
wan             | 58 (5)            |              | +    |
| Income                                                                                                                                     | Annual dispos
household inco
(1000 NT\$)      | able
omes                       | DGBAST b ,          | Taiwan                 | 660.1 (2          | 23.7)        | _    |

| Employment        | Employed              | DGBAST, Taiwan | 0.15 (0.26) | _ |
|-------------------|-----------------------|----------------|-------------|---|
|                   | population(employed   |                |             |   |
|                   | population/           |                |             |   |
|                   | population)           |                |             |   |
| Production values | Annual production     | DGBAST, Taiwan | 27.9 (81.4) | _ |
|                   | values of industries  |                |             |   |
|                   | and services (million |                |             |   |
|                   |                       |                |             |   |

|                     | NT\$)                                                                                                                                    |                                                                        |                                                                        |                                     |         |            |   |
|---------------------|------------------------------------------------------------------------------------------------------------------------------------------|------------------------------------------------------------------------|------------------------------------------------------------------------|-------------------------------------|---------|------------|---|
| Land uses           | Urban
developments                                                                                                                    | Area of
residen
comme
industr
educati
public 2
(km2) | f
tial,
crcial,
ial,
onal and
land uses                 | Land Use
Investigation
Taiwan | n of    | 0.35(0.48) | + |
| Agricultural uses   | Areasof agricultural
land uses (km2)
Environmentalsensiti
ve areas (km2), e.g.,
flood plain, mountain
slope reserve areas |                                                                        | Land Use Investigation2.17(2.91)of Taiwan10.4(40.4)of Taiwan10.4(40.4) |                                     |         | .91)       | + |
| Sensitive areas     |                                                                                                                                          |                                                                        |                                                                        |                                     |         | 0.4)       | _ |
| Road infrastructure | Areas of road infrastructure                                                                                                             | (km2)                                                                  | Ministry o
Interior, Ta                                             | f the
aiwan                      | 0.16(0. | .14)       | + |
| Adaptive capacity   | Shelters                                                                                                                                 | Numbe                                                                  | er of
S                                                             | Measured b                          | oy GIS  | 1.16(1.43) | _ |
| Fire and police     | Number of fire                                                                                                                           | e and
ver                                                           | Countyand                                                              | city                                | 2.05(2. | .02)       | _ |
| Medical services    | Hospital beds                                                                                                                            | , ci                                                                   | Countyand                                                              | l city
nt                        | 10.5(10 | 6.0)       | _ |
| Risk perceptions    | Average levels
perceived resid
risk to climate
hazards (5-poi
Likert scale)                                                  | s of
dential
nt                                                  | Questionna
interviews                                               | aire                                | 2.97(0. | 17)        | _ |
| Access to resources | Average levels
ability to access
resources (5-p
Likert scale)                                                                   | s of
ss to
oint                                                  | Questionn:
interviews                                               | aire                                | 2.03(0. | .18)       | _ |
| Adaptation appraisa |  <li>Average level:
residents evalu
their ability to
perform adapt
successfully (
Likert scale)</li>        | s of
late
ations
5-point                                      | Questionna
interviews                                               | aire                                | 2.43(0. | .50)       | _ |

---

## Referee Comment (RC2) · Anonymous Referee #2 · 26 Jul 2016

Review for the manuscript "Linking local vulnerability assessments to climatic hazard losses for the river basin management"

The authors present a study that seeks to link typhoon losses within three river basins with vulnerability and coping capacity/adaptive capacity indicators. Particularly, the authors try to determine the statistical relationships between selected indicators and the observed losses using various regression methods. The presented study could provide interesting empirical/quantitative insights into the usefulness and the effects of, e.g., mitigation methods in the context of risk management, and might thus be considered for publication. There are, however, numerous substantial issues that need to be addressed by the authors, primarily in regard to the underlying conceptual framework.

[Figure]

In the following, I am listing my main concerns. Despite those, the manuscript requires substantial language editing. I have just listed selected instances where language editing is required.

(*) The authors state their aim to "examine whether geographic localities characterized by high vulnerability experience significantly more damage [...]" (p. 3, l. 5). Significantly more damage in comparison to what? Please rephrase and clarify.

(*) It is argued that several events hit the studied river basins. Maybe provide some more examples on additional events to put the study into a broader context, since just a singular event, i.e., Morakot, surely would not trigger exhaustive risk management but might be more treated like an outlier.

(*) P. 4, l. 2: I feel like you need more detail/consistency in your approach when it comes to the usage of terms, to better describe your conceptual framework (this is a comment also in light of the vague definition of vulnerability, please see a comment on that below). Here, what are biophysical elements, socioeconomic elements, industrial elements (aren't these belonging to socio*economic* elements?), "and land use elements" (sic!). Don't all these elements constitute some form of land-use?

(*) P. 4, l. 5: What is meant with properties of a specific watershed context?

(*) P. 4, l. 11: The definition of vulnerability that is used by the authors remains vague. Please elaborate more on the concepts of exposure, sensitivity, adaptive capacity employed in your framework as shown in Eq. 1. I take it that, here, exposure refers to the UNISDR definition, i.e., elements potentially at risk? Also, later on, in Eq. 3, you refer to loss as a function of hazard and vulnerability. Thus, following your argumentation, e.g., I do not see averaged annual rainfall or debris flow as elements of the entity exposure (and, therefore, vulnerability), but rather as entities of the hazard (or hazard magnitude) used to determine affected (exposed) area/elements. I would also argue on the indicators elevation and proximity to rivers, which I do not see as an indicator for sensitivity. In order to clarify such issues, please elaborate in more detail on your conceptual

framework/the definitions employed in your study. Also, if you argue that vulnerability frameworks need to be integrated, why didn't you investigate into additional, integrated frameworks of vulnerability?

(*) In doing so, please also provide more insight into why the criteria listed in table 1 have been selected/how selection has been done.

(*) P. 7, l. 18: It could be argued if increasing income etc. enhances coping strategies so that vulnerability is reduced, since an increase or accumulation of wealth etc. is typically also seen to increase vulnerability by increasing the values potentially at risk.

(*) P. 7, l. 23: If you argue that vulnerability is a multi-dimensional concept (which is not defined in the manuscript), I do not see how a preservation of (environmentally) sensitive (i.e., vulnerable?) areas leads to a decrease in vulnerability?

(*) P. 6, l. 12: What are the synergies and complexities that have been obtained and discussed by Hung & Chen? What is their relevance for the presented study? What is the "framework of vulnerability indicators" that is being referred to? As mentioned above, please elaborate more on your methodology and conceptual frame.

(*) In regard to Eq. 3, it is argued that using just one event (i.e., the typhoon Morakot) allows for the control of the "disaster scenario", so that variation in loss can be attributed to variation in vulnerability. How do you control for hazard intensity as a governing factor of loss? Again, please provide more information on the methodology.

(*) I have the impression that also the term hazard impact needs more clarification. Following table 1, hazard impact seems to refer to casualties, losses etc. However, in the manuscript, p. 9, l. 13, it is argued that the interaction of hazard impacts and vulnerability generates loss. In this case, wouldn't impact refer to hazard intensity?

(*) In table 1, you list various indicators per category. Please, describe what is the Mean/standard deviation referring to? Is, e.g., mean of population density equal to 2.74 inhabitants/$km^2$/village unit? Isn't this number rather low, also considering your

argumentation that the three river basins have approximately 1.26 mio. inhabitants and an area of 7885km$^2$ (which would equate to approximately 160 inh./km$^2$)?

(*) Please provide more information on your indicators used to assess adaptive capacity. E.g., you mention the indicator "access to resources". What is meant with this indicator? What resources, access in which way, etc., could you please provide more detail? How is the indicator operationalized? Also in this regard, what is the number of interviews in total, and what is N per river basin?

(*) Please elaborate more on the approaches that you classify as "top-down" and "bottom-up".

(*) Also, in regard to both approaches, you argue on the importance of connecting them, and on their "relative magnitude" (p. 5, l. 21). What is meant with that, what is relative magnitude? Furthermore, do you see both types of approaches as distinct, since you argue for a further integration of both? Isn't it the case, however, that "top-down" approaches also make use of empirical data (losses etc.) for validation of models, hence, integrate both approaches?

(*) Generally, please provide more details on your statistical analysis. I suggest you to include e.g. results as supplementary material if possible, e.g., correlation coefficients etc. I assume that your analysis is carried out per river basin and per village unit, i.e., non-spatial?

(*) I find it difficult to interpret Spearman/Pearson r without knowledge on the shape of the distribution of indicators $\sim$ losses, as e.g. Pearson would only make sense in linear relationships. Do you find linearity? Again, what is N per river basin?

(*) Abstract, l. 8: Please rephrase "attack". (*) Introduction, P. 1, l. 24: Remove meanwhile. It sounds like 700 people were killed during the typhoon due to something completely different. (*) P. 5, ll. 12-18: The paragraph is unclear to me. (*) P. 10, l. 1.: Please rephrase, use e.g. people or inhabitants instead of populations. (*) P. 10, l. 18:

[Figure]

I guess you mean administrative instead of geopolitical boundaries? (*) P. 11, l. 9: Do you mean spatially defined clusters? (*) P. 12, l. 13: 93% of typhoon Morakot caused no damage or injuries. This sounds a bit odd.
* * *

---

## Author Comment (AC1) · 5 Sep 2016

Replies to the interactive comment on "Linking local vulnerability assessments to climatic hazard losses for river basin management" Anonymous Referee #1

The authors would like to sincerely acknowledge the anonymous referee for providing us very valuable suggestions and comments. In particular, the referee's suggestions are really helpful for improving our article's conceptual framework, language and overall quality. We summarize the replies and relevant ways of dealing with each comment or suggestion as follows.

1. This paper deals with the losses and damages of a typhoon and relates these to

[Figure]

a composite vulnerability index. The underlying datasets and the statistical work is of relevance for the scientific community. However, there are significant lacks in - The conceptual frame for the work - A critical view of the approach followed and of the results achieved. *Replies: To improve the expression of our conceptual frame, the authors will rewrite the section 2 (Vulnerability and disaster impacts) of the revised manuscript to elaborate more on the process and basis of building our conceptual framework. Moreover, in the section 3 (Methods and data), we will also increase a paragraph and a conceptual figure (Figure 1) to help explain our basic concept and the stepwise procedure of analyses. In the manuscript, it really needs more discussions about the approaches followed and the findings achieved. Thus, in section 2, the authors will add more sentences to explain their vulnerability concept and the approaches followed, as well as to describe more on the linkages between vulnerability and disaster losses. In addition, in the section 4.4 (Policy implications) and section 5 (Conclusions), we will reinforce the explanations about our findings and the limitations in our approaches.

2. In addition, the text has to be significantly improved regarding the English language. I have given a number of proposed corrections in the first half of the text (See below). *Replies: We really appreciate all the help the referee give us. It is greatly helpful for us to improve our writing and to clarify some basic concepts in our manuscript. To improve the language problem in this manuscript, besides the referee's suggestions (please see the attached rudimentary revised manuscript), we will reorganize the whole manuscript, including the word choice, grammar, sentence usage and structure. Finally, the revised manuscript will also have a thorough edit for making the content more intelligible and more suitable for publication.

3. I would encourage the authors to review their paper thoroughly and particularly regarding the various concepts of vulnerability. I would also like to ask them to take the constraints of their methodologies into consideration when discussing their results. *Replies: In the revised manuscript, the authors will reinforce the discussions about various concepts of vulnerability. This could help the authors elaborate the process of

developing their conceptual vulnerability framework. On the other hand, we will also increase the discussions about the constraints of our methodologies in explaining the findings of our case study.

4. The concept of vulnerability and the implication that the conceptual approach has on the study is not clear: - At the end of chapter 2.1 the authors state that it is necessary to integrate the vulnerability concepts of the disaster community and of the IPCC. The proposed formular (1) however does reflect only the IPCC concept. If the authors start to discuss these conceptual issues they need to be much more sharpened in their explanation of the differences of the various approaches and why and how they would like to integrate approaches. *Replies: Indeed, in the manuscript, the explanations about the process of our conceptual vulnerability framework was not very clear. In the revised manuscript, the authors will reinforce the explanation about how we conceptualize vulnerability. We will focus more on the descriptions of how we combine the IPCC concept and risk-hazard approach (particularly, UNISDR concept) to generate the formula (1) and formula (2). Especially, the formula (2) is built based on the risk-hazard approach that allows us to link the concept of vulnerability to hazard losses. In addition, we will also strengthen our discussions on why and how we assess vulnerability and examine its relations with hazard losses using an integrated approach. We argue this issue are mainly based on the requirements of integrated river basin management.

5. The two approaches for investigating the relationships between vulnerability and disaster losses in chap 2.2 are not described clearly enough. *Replies: In the section (chapter) 2.2 (Vulnerability and disaster losses) and 3.1 (Linking vulnerability factors and climatic disaster losses) of the revised manuscript, the authors will increase more explanations about the relationship between risk-hazard and IPCC approaches to vulnerability and disaster losses. In the section 2.2, we classify existing studies into two types. One type of approach focuses on combining existing hazard loss theories (such as risk-hazard, PAR, PSR theory or MCDA) with computer-aid simulation and GIS-based analysis to predict disaster losses, while another type of approach empha-

sizes on applying existing or surveyed databases to characterize the distributions of disaster damages. In the section 3.1 (Linking vulnerability factors and climatic disaster losses), we will further add a conceptual framework to link above mentioned two types of approaches, as well as to explain how we apply this framework to develop our methodology and analytical procedure.

6. The methodology for the selection of indicators is not transparent. There is a lack of clarity in the concept reflected in the description of the indicators in chapters. 3.1.X. For example, coping is mentioned as part of both sensitivity and adaptive capacity. *Replies: In the original manuscript, it really lacks clarification of the methodology of our indicator selection. Thus, in the revised manuscript, the authors will further provide more explanations about the basis and process of indicator selection. In reality, the procedure of indicator selection are stemmed from our conceptual vulnerability framework, as well as from a summarizing review of the literature and the contextual characteristics of the river basin management. Moreover, we will reorganize the classification of the indicators in order to more conform with their characteristics.

7. A critical reflection on the selection of a limited number of indicators is missing. *Replies: Indeed, the authors should discuss the constrictions of the selection of a limited number of indicators to assess vulnerability and to examine its relations to disaster losses. In the revised manuscript, the authors will discuss more on the limitations of their studies, while they describe the implications and applications of their findings to supporting the process of policy-making.

8. A discussion of the problems when using statistical methods when only limited damage and loss data is available is entirely missing. *Replies: In the revised manuscript, the authors will enhance the discussions about the restrictions of using limited typhoon damage and loss data to examine the relationships between vulnerability and disaster losses. In particular, the findings are based on a single hazard event approach, which would limit their applications in other hazardous events. Thus, our study focuses more on developing a conceptual framework and methodology of examining the relationships

between vulnerability and hazard losses. These related discussions will be reinforced in the descriptions of our findings and their implications.

9. A description of the typhoon event itself is missing. *Replies: The authors will give a brief description about Typhoon Morakot. This typhoon, a Category 1 typhoon, hit southern Taiwan during 8–12 August 2009. It was the most severely damaged typhoon in Taiwan in the past 50 years. This typhoon caused torrential rainfall that resulted in widespread flooding and thousands of landslides. Typhoon Morakot killed nearly 700 people and left thousands of people either displaced or homeless. The estimated total amount of economic losses was approximately US$ 0.6 billion. These descriptions about Typhoon Morakot will be added in the revised manuscript.

10. It is not clear for which spatial extend the regression analysis has been carried out. For example, what was the spatial resolution of rainfall data? How did the authors deal with the fact that the data is available in different formats (point, raster etc). *Replies: The data unit used in our regression analyses is a village. This spatial resolution of data is also available for extending to various formats, such as point, raster, etc. But in practice, the survey values of various data could contain different scales and units. It still needs some processes of data processing to fit the requirements of regression analysis. For example, we used an average annual rainfall value (the past ten years) surveyed by the nearest rainfall station (set up by the Central Weather Bureau, Taiwan) as the rainfall data for each village. In the revised manuscript, we will also increase the related discussions about the potential of extending our methodology to other areas and other formats of data.

11. The MCDA has not been described in detail, what is it exactly and which role does it play? *Replies: In the manuscript, it is really rarely described in detail about the role of the MCDA. To improve the descriptions, in the revised manuscript, we will add an explanation about how we develop the conceptual framework of vulnerability and apply this framework to assess integrated vulnerability over the three case study river basins using an MCDA procedure.

12. The discussion needs to consider the problem to look at hazard and vulnerability factors separately. The authors state that "villages with higher elevations, in upper streams and more proximity to rivers tended to suffer more disaster casualties and losses due to their higher exposure to typhoon impacts". Unclear remains what the difference is between exposure and typhoon impacts (are impacts = damage?). Then they conclude, "However, constraints associated with local government adaptation efforts in the river basins reflect a range of challenges in relation to how the integrated RBM adaptation efforts have structured. The efforts to facilitate adaptation should largely target the mitigation of vulnerability and risk." – these types of conclusions need to be explained further. *Replies: Many thanks for the referee's valuable suggestions. Indeed, in the manuscript, there is a bit confusing in the distinction between the terms of "exposure" and "hazard impacts". In our study, we consider the disaster risk/loss as a function of hazard and vulnerability. In a long-term scale, disaster risk requires the consideration of uncertainties in both hazard and vulnerability factors due to ambiguities in the possible changing of the future hazard and vulnerability factors over time. However, in a short-term or single hazard event scale, it is a deterministic approach, which the focus is on a single disaster loss that could vary with different hazard intensities, impacts and vulnerability factors. In our study, we applied a single hazard event approach. We also consider "exposure" as a factor of vulnerability, which refers to the presence of areas, system or assets in places and settings that could be adversely affected. The term "hazard impacts" involves the hazard intensities and their effects on the areas, which could include the hazard damages and its physical influences (e.g., flooding, landslides) on the areas. The distinction between the concepts of these two terms is really not clear in the manuscript. We will clarify these differences by increasing more explanations about the concepts of these two terms in the revised manuscript. Furthermore, in the conclusions, we will rewrite, reorganize and increase more sentences in the section 4.4 (Policy implications) in order to more clarify the related discussions about the policy implications of our findings.

Please also note the supplement to this comment:
http://www.nat-hazards-earth-syst-sci-discuss.net/nhess-2016-114/nhess-2016-114-
AC1-supplement.pdf
* * *
Interactive
comment

[Figure]

[Figure]

**Figure 1.** Stepwise procedure and framework of analysis

**Supplement:**

**Linking local vulnerability assessments to climatic hazard losses for river basin management**

Hung-Chih Hung1, Yi-Chung Liu2, Sung-Ying Chien2

1 Department of Real Estate and Built Environment, National Taipei University, New Taipei City, 23741, Taiwan
 2 National Science and Technology Center for Disaster Reduction, New Taipei City, 23143, Taiwan

Correspondence to: Hung-Chih Hung (hung@mail.ntpu.edu.tw)

Abstract. To prepare for and confront the potential impact of climate change and related hazards, many countries have implemented programs of integrated river basin management. This has led to an imperative challenge for local authorities to improve the understanding of how the vulnerability factors link to climatic disaster losses. This article aims to examine

- 10 whether highly vulnerable areas experience significantly serious damage caused by weather extreme events at the river basin levels, and explain what vulnerability and hazard impact factors determine the disaster losses. Using three river basins in southern Taiwan attacked by Typhoon Morakot in 2009 as case study areas, we proposed a novel methodology that combines a geographical information system (GIS) technique with a multicriteria decision analysis (MCDA) to evaluate and map composite vulnerability to climatic hazards across river basins. Then, the linkages between hazard impacts,
- 15 vulnerability factors and disaster losses are tested by using a disaster damage model (DDM). In the case study, the results of the vulnerability assessments indicated that the vast majority of the most vulnerable areas is situated in the regions of the middle, and upper reaches and some coastlines of the three river basins. Using the DDM, it shows that the losses and casualties due to typhoon are significantly affected by local vulnerability contexts and hazard impact factors. Finally, we suggest the implications of adaptation policy lines for minimizing vulnerability and risk and for integrated river basin
- 20 governance.

25

5

**1 Introduction**

Major portions of Asia have an increasing exposure and vulnerability to climate change and weather extremes due to rapid urbanization and overdevelopment in hazard-prone areas (IPCC, 2014). Particularly, Asia-Pacific region is the riskiest and the most seriously affected areas of the world. More than 1.2 billion people have been exposed to climate-related (climatic) hazards, and the number of people residing in cyclone-prone areas has grown from 71.8 million to 120.7 million (UNISDR.) 註解 [H1]: Replies to the Comment [S1]: We have rewritten this sentence. 註解 [H2]: Replies to the Comment [S2]: We have checked the language

problem.

註解 [H3]: Replies to the Comment [S3]: We have reworded this phrase.

註解 [H4]: Replies to the Comment [S4]: We have added more explanations in this sentence.

註解 [H5]: Replies to the Comment [S5]: The authors have checked the language problem.

註解 [H6]: Replies to the Comment [S5]: We have reorganized this paragraph and add more descriptions about recent trends of disaster impacts. 2012). Thus, it becomes increasingly important for water resource managers to implement integrated river basin management (RBM) that can cope with and reduce the potential impacts of climate change and climatic disaster risks (Hung et al., 2013).

Integrated water resource management is a process to promote the coordinated development and management of water, land uses and related resources (GWP 2000). This indicates that the integrated RBM program should adopt the river basin as a management unit, employing a comprehensive perspective to connect water resource management, agricultural irrigation with land use planning for building more resilient river basin contexts (Penning-Rowsell et al., 2006). Especially, vulnerability assessment plays a vital role for decision makers in scrutinizing the biophysical and socioeconomic conditions, as well as their distributions over river basins. This process of assessment also helps decision makers integrate various local

5

10 connections into planning and policy lines for disaster damage and risk mitigation within the context of whole river basins (Hooijer et al., 2004; Hung and Chen, 2013; You and Zhang, 2015).

Climatic hazard losses and risk accumulation result from the interlinking between hazard and vulnerability factors, which has led to an emerging literature focus on characterizing these factors and their interaction (UNISDR, 2012; Hung et al., 2013). Within these studies, existing vulnerability analyses majorly focused on assessing, mapping and distinguishing the

- 15 variability of the vulnerability distribution between regions (Adger, 2006; Hung and Chen, 2013; Ahumada-Cervantes et al., 2015). Most previous disaster loss and risk studies use computer-aided simulation, scenario analyses and multicriteria decision analysis (MCDA) (Tate et al., 2010; Ni et al., 2010; Hung et al., 2013; De Bruijn et al., 2014). Their findings are valuable in characterizing disaster risk, impacts and distributions that enable decision makers to create risk maps and communicate the high risk areas to stakeholders. Few studies have systematically examined how the vulnerability and hazard
- 20 impact factors are linked to their potential effects on disaster losses (Hung and Chen, 2013; Visser et al., 2014). This would compromise the application of existing vulnerability and exposure studies to the disaster risk assessment and integrated RBM.

This article aims to examine whether localities characterized by high vulnerability experience significantly higher damage than other areas owing to onset weather extreme events at the river basin level, and explain what vulnerability, hazard and exposure factors influence these damages or losses. Using three river basins in southern Taiwan hit by Typhoon Morakot as case study areas, we propose a novel methodology based on existing disaster impact theory, which then combined an MCDA, 註解 [H7]: Replies to the Comment [S7]: We have reworded this sentence.

**註解 [H8]:** Replies to the Comment [S9]: The authors have deleted the term "however".

**註解 [H9]:** Replies to the Comment [S10]: We have rephrased this sentence.

註解 [H10]: Replies to the Comment [S8]: We have checked the language problem and reformulate this sentence.

註解 [H11]: Replies to the Comment [S11]: We have replaced the term "extant" with "previous".

註解 [H12]: Replies to the Comment [S12]: We have reworded this phrase.

註解 [H13]: Replies to the Comment [S13]: The authors have added some references.

註解 [H14]: Replies to the Comment [S13]: We have checked the language problem and reworded it. GIS (geographical information system)-based statistics with multivariate analysis to assess climatic hazard vulnerability (especially typhoons and floods). Moreover, we examine the connection between vulnerability, hazard impact factors and disaster losses using a disaster damage model (DDM). This methodology may also be applicable to other river basins. Finally, we discuss the extension of our findings in providing policy directions for building adaptive capacity and for integrated RBM.

**2 Vulnerability and disaster impacts**

5

25

**2.1 Vulnerability and its assessment**

Vulnerability assessments have been broadly applied to various research communities with respect to climate change adaptation and disaster risk management, although not agreeing on a common view about the concept of vulnerability. In the

- 10 traditions of disaster risk research, risk-hazard approach describes vulnerability as the degrees of susceptibility of these assets to suffer damage and loss (UNISDR, 2013). It denotes the relationships between the expected damages and the sensitivity and exposure attributes of the affected systems (Füssel, 2007). The disaster pressure-and-release (PAR) and pressure-state-response (PSR) models take this vulnerability concept as a starting point, defining risk as the product of hazard and vulnerability (Wisner et al., 2004).
- 15 Existing applications of the risk-hazard approach emphasized on mono-dimensional analyses, which were either focused on engineering, biophysical or socioeconomic vulnerability assessments (Adger, 2006; Mokrech et al., 2012). On the other hand, some studies had extended their applications in different integrated approaches, most notably the hazard-of-place model (Cutter et al., 2000). IPCC (2014) had conceptualized vulnerability as which encompasses a variety of concepts and elements, including sensitivity or susceptibility to harm and lack of capacity to cope and adapt. Such definition of
- vulnerability comprises adaptation capacity, which is referred to as the end point view of vulnerability (O'Brien et al., 2007).
   It, therefore, provides an integrated concept that links starting point and end point view of vulnerability (Adger, 2006; Scheuer et al., 2011).

From the perspective of integrated RBM, it needs transdisciplinary, multi-dimensional and more inclusive vulnerability approaches to involve various biophysical and socioeconomic properties of a specific river basin area in the risk management. Thus, vulnerability assessment should facilitate decision-makers to engage in the integrated analyses of the interaction between the components of vulnerability and the properties of a specific river basin context (O'Brien et al., 2007; Engle and Lemos, 2010; Hung and Chen, 2013). To target support for integrated RBM, it needs to integrate IPCC (2014) with risk-hazard approaches to build more transdisciplinary and comprehensive vulnerability assessment framework. Therefore, the vulnerability can be generally described as a function of exposure, sensitivity and adaptive capacity:

(1)

5

25

**2.2 Vulnerability and disaster losses**

Vulnerability = f (exposure, sensitivity, adaptive capacity)

Existing approaches to the relationships between vulnerability and disaster losses can be divided into two major types. The first type of approach frequently uses risk-hazard, PAR, PSR theory or MCDA with computer-aid simulation and GIS-based analysis to predict disaster losses (Ermoliev et al., 2000; Wisner et al., 2004; Tate et al., 2010; Scheuer et al., 2011; De Bruijn

- 10 et al., 2014). PSR and PAR approaches show the linkage between hazard event and unsafe context that leads to disaster, as well as present how vulnerability influence hazard loss and risk (Wisner et al., 2004). They mostly take a risk-hazard approach to estimate expected damages caused by various kinds of hazards, which assumes that disaster risk is composed of two factors: hazard and vulnerability (Füssel, 2007). In a long-term scale, disaster risk requires the consideration of uncertainties in both hazard and vulnerability factors due to ambiguities in the possible changing of the future hazard and
- 15 vulnerability factors over time (De Bruijn et al., 2014). However, in a short-term or single hazard event scale, the focus is on a single disaster damage that could vary with different hazard intensities, impacts and vulnerability factors (Cutter et al., 2003; Hung et al., 2013; Hung and Chen, 2013). This type of study majorly combines computer-aid approaches with MCDA in modelling or mapping vulnerability and risk (Scheuer et al., 2011; Hung and Chen, 2013). Moreover, most these studies use current vulnerability factors to project future risks or damages, which is a 'top-down' approach that can bring predicted
- 20 distributions of disaster impacts and risks to the fore.

The second type of approach focuses on 'bottom-up' and data-based analysis, which often uses existing or surveyed databases to characterize the distributions of disaster damage (Zahran et al., 2008; Bhattarai et al., 2015). The findings not only can help decision makers identify disaster loss distributions, but also understand their determinants (Downton and Pielke, 2005). This approach concentrates more on mapping the disaster damage distribution at the national or regional levels than the local levels. It also majorly combines expert judgment with mono-dimensional evaluation to inspect the influential

註解 [H15]: Replies to the Comment [S15]: The authors have checked the language problem and added some descriptions in this sentence.

註解 [H16]: Replies to the Comment [S18]: We have rephrased this sentence for more clarifying our descriptions. 註解 [H17]: Replies to the Comment [S16]: We have added more descriptions about how the risk-hazard approach conceptualizes hazard risk. 註解 [H18]: Replies to the Comment [S16]: The authors have rephrased this sentence and added some references.

註解 [H19]: Replies to the Comment [S19]: We have reworded this phrase and added more explanations in the following paragraph.

**註解 [H20]:** Replies to the Comment [S20]: We have reworded this sentence for more clarifying the explanations.

factors of disaster damage (Mokrech et al., 2012; Hung and Chen, 2013). However, little attention had been paid to linking of a multi-dimensional vulnerability assessment with an empirically-based disaster loss evaluation in the river basin contexts. As above-mentioned, the first type approach seeks to systematically identify disaster losses and scrutinize their

components, as well to project various disaster impacts resulting from different hypothetical events. By contrast, the second type approach enables decision makers a conjoint treatment of quantitative disaster loss data and qualitative human judgment. Nonetheless, these two types of approaches all considering disaster losses are inherent and dynamic due to ongoing

interaction of climatic hazard impacts with the biophysical and socioeconomic components of vulnerability in a watershed

system (O'Brien et al., 2007; Maru et al., 2014).
Increasing the understanding of the formation of climatic disaster risk highlights the importance of connecting
aforementioned two types of approaches and their relative magnitudes in hazard risk analyses (Mokrech et al., 2012; Visser et al., 2014). Particularly, incorporating the first type into the second type approach allows us to create frameworks of

**et al., 2014). Particularly, incorporating the first type into the second type approach allows us to create frameworks of disaster risk analysis that could assist in expanding the range of vulnerability assessments and in sequencing them to generate robust resilience and adaptation pathways (Hung et al., 2016).**

**3 Methods and data**

5

- An MCDA and GIS-based statistic analysis is integrated with a data-based multivariate analysis, in order to assess and map the vulnerability of three river basins in southern Taiwan and to examine the vulnerability factors that influence the disaster loss distributions. [The procedure of analysis consists of three steps (Fig. 1)]. First, using the vulnerability defined by equation (1), we constructed a composite vulnerability indicator framework and combined with an MCDA to assess and map climatic hazard vulnerability at the river basin level.] Second, based on the risk-hazard framework, the relationship between disaster
- 20 loss distributions, impacts and vulnerability factors was tested and compared using numerous regression models. Finally, we discussed the findings and provided implications for better adaptation policy lines.

(Figure 1. Stepwise procedure for linking local vulnerability assessments to hazard loss analyses)

**3.1 Linking vulnerability factors and climatic disaster losses**

This study focuses on a single disaster event scenario for comparatively static modelling of hazard damages or losses at

註解 [H21]: Replies to the Comment [S21]: The authors have added more descriptions about the differences between "mono-dimensional" and "multi-dimensional" vulnerability assessments in this section (section 2.2).

註解 [H22]: Replies to the Comment [S22]: We have reworded this phrase.

註解 [H23]: Replies to the Comment [S23]: We have reworded this phrase.

註解 [H24]: Replies to the Comment [S24]: We have rewritten this sentence. 註解 [H25]: Replies to the Comment [S26]: The authors have added a Figure to explain their conceptual framework (including how we combine the risk-hazard approach with IPCC's concept) and the stepwise procedure for conducting their empirical study.

註解 [H26]: Replies to the Comment [S25]: The authors have rephrased this sentence for more clarifying their approach to assessing vulnerability. different points over river basins. This approach allows us to concentrate on single disaster scenario, so that any variation in losses can be directly resulted from changes in hazard impacts and vulnerability factors (Hung et al., 2013). Therefore, the disaster loss model can be written as the following function:

(2)

Disaster loss/risk = f (hazard, vulnerability)

5

Equation (2) implies that disaster loss/risk is a function of hazard and vulnerability factors. In a short-term or a single disaster event scenario, the extent of disaster loss varies with hazard impacts (or intensities) and vulnerability factors. To more specifically identify the relationship between disaster losses and vulnerability factors, several regression models were used in the case studies.

**3.2 Indicators of the vulnerability framework and hypotheses**

- 10 An indicator-based assessment framework was developed with the aim to identify composite indicators that can serve as proxies for the components of vulnerability. Vulnerability assessments and mapping have widely used indicator-based approach combined with GIS to help stakeholders characterize distributions of vulnerable areas and understand factors leading to vulnerability (Cutter et al., 2003; Hung and Chen, 2013; Ahumada-Cervantes et al., 2015). The vulnerability assessment framework created here allows us to take advantage of the contributions of existing knowledge, as well as
- 15 obtains the synergies and complexities of watershed contexts as discussed in detail in Hung and Chen (2013). According to our conceptual vulnerability, the indicators involved in the assessment framework consist of three dimensions: exposure, sensitivity and adaptive capacity. The indicators are selected stemed from a summarizing review of the literature and the contextual characteristics of the river basins in southern Taiwan.

In terms of assessing the integrated vulnerability, we mainly adopted the framework of vulnerability indicators 20 promulgated by Hung and Chen (2013), which was appropriate and widely applied to the river basin conditions in Taiwan. Hung and Chen (2013) also identified vulnerability based on the concept of IPCC. It had applied focus group meetings and in-depth interviews of experts, officers and community members to incorporate key stakeholders' participation and knowledge into an MCDA procedure. Then, an assessment of composite vulnerability was conducted across the case study areas at the village scale, which is the basic unit of local administration in Taiwan. Finally, those indicators considered to assess vulnerability are demonstrated in Table 1, along with their descriptions, data sources and the expected direction of the 註解 [H27]: Replies to the Comment [S35]: The authors have increased some explanations about how they conceptualized the disaster loss/risk in their study.

註解 [H28]: Replies to the Comment [S27]: The authors have added more explanation about why they employ an indicator-based approach.

註解 [H29]: Replies to the Comment [S28]: We have increased a bit of descriptions about the Hung and Chen's (2013) study. relationship to disaster losses.

(Table 1. Hazard impacts, vulnerability indicators (variables) and expected sign to disaster losses)

**3.2.1 Exposure indicators**

Exposure refers to the presence of areas, system or assets in places and settings that could be adversely affected (IPCC, 2014;

- 5 Hung et al., 2016). To reflect the degrees of exposure, *averaged annual rainfall* and *potential debris flow torrents* were used. The expectation is that either higher rainfall or greater debris flow torrents enhance vulnerability and thus enhance the likely disaster losses1 (Scheuer et al., 2011). Furthermore, the biophysical contexts also can be used to measure the extent of an area exposed to hazards. The hypothesized links between biophysical contexts and disaster losses are captured in examining the influence of *proximity to rivers* and *elevation* indicators on disaster losses. The areas where are more proximity to rivers
- 10 and/or at higher elevations are more sensitive and vulnerable, which could increase disaster damages (Ni et al., 2010)

**3.2.2 Sensitivity indicators**

Sensitivity is a one of the most broadly used attributes to describe the vulnerability in climate change and disaster risk management (Cutter et al., 2003; O'Brien et al., 2014). The sensitivity indicators are mostly composed of inherent socioeconomic and land use sensitivity (Hung and Chen, 2013). The socioeconomic indicators include *populations*, *social*

- 15 dependence, income, employment and production values of industries and services. These indicators are apt to reflect the extent of areas' contextual vulnerability in a watershed. Thus, increasing income, employment and/or production values by communities is expected to enhance coping strategies, thereby decreasing vulnerability and potential disaster losses (Zahran et al., 2008). Contrarily, populations and social dependence have expected a positive relation to disaster damage (Hung et al., 2016).
- 20 In the aspect of land-use, the indicators comprise *urban developments, agricultural uses, environmentally sensitive areas* and *road infrastructures.* Generally, while preserving more sensitive areas could decrease vulnerability and disaster losses,

**註解 [H30]:** Replies to the Comment [S29]: We have replaced the reference "Wisner et al. (2004)" with "Scheuer et al. (2011)".

註解 [U31]: Replies to the Comment [S31]: We have reorganized the classification of the indicators. Particularly, the indicators of biophysical contexts have re-categorized as components of "exposure" dimension.

註解 [H32]: Replies to the Comment [S32]: We have deleted the term "fragility".

<sup>1 The *averaged annual rainfall* can be also considered as an indicator of climatic hazards. However, in the long-term scales, we deemed it as a measurement of the levels of an area exposure to climate hazard because the areas with higher averaged rainfall could have higher probability of exposure to hazards.

the larger scales of either urban development, agricultural use or road infrastructure would encourage denser land use, agricultural developments and tourist activities, and that could lead to higher vulnerability and expected disaster losses (Cutter et al., 2003; Mehaffey et al., 2008).

**3.2.3 Adaptive capacity indicators**

5 Using adaptive capacity indicators to measure the ability of communities to adjust to potential damage, to take advantage of opportunities, or to respond to disaster consequences (IPCC, 2014), the indicators include *shelters, medical, fire and police* services. These indicators present an area's abilities of coping, evacuation and emergency responses. Therefore, improving these facilities could reduce vulnerability and likely disaster damage. We also involved behavior and heuristic factors that consisted of residents' *risk perceptions*, their ability to *access resources* and to successfully adapt to hazards (self-efficacy) are also considered. The hypothetical relationships between these factors and disaster damage are negative (Eakin et al., 2010).

**3.3 Composite vulnerability index**

15

To assess the integrated vulnerability for each village, the composite vulnerability index (CVI) was estimated using an MCDA procedure. This procedure comprises three steps. First, because the survey values of various indicators shown in Table 1 contain different scales and units, we applied a min-max scaling to directly normalize all of the data into a uniform [0, 1] scale with ratio properties. Second, the normalized values are then used to compute CVIs by:

(3)

$$CVI_i = \sum_{j=1}^m w_j x_{ij.}$$

where  $CVI_i$  represents the composite vulnerability index for village *i*;  $x_{ij}$  denotes the normalized value for indicator *j* and  $w_j$  is the weight. With above hypotheses, if  $x_{ij} > 0$ , it indicates higher levels of overall vulnerability; if  $x_{ij} < 0$ , decreasing or lessening the overall vulnerability. Third, equal-weight was assigned to each indicator in order to build an equivalent basis for comparing the attributes of vulnerability and disaster losses among the river basins.

註解 [H33]: Replies to the Comment [S34]: We have added a bit of explanations about the CVI, as well as reorganized the related descriptions in this sentence.

註解 [H34]: Replies to the Comment [S33]: The authors have reorganized and rephrased the related descriptions in this sentence.

**3.4 Case study areas and data**

This article explores three very different river basins, choosing for representing the areas with various degrees of development and different contexts in southern Taiwan (Fig. 2), but all having heavily struck by Typhoon Morakot in 2009. The case study areas include three major river basins: Gaoping, Tsengwen and Taimali River. According to the 2015 census,

- 5 these three river basins encompass 598 villages, around 1.26 million inhabitants and cover an area of approximately 7,885km2. Highly diversified topography distributes over the three watersheds. The altitude of this region ranges from coastal lowlands along the western shoreline to above 3,000 meters in the eastern high-mountain areas. Uncontrolled urban sprawl and environmental destruction are interwoven by growing threats from climate change and weather extremes mean that lead to the riskiest regions in Taiwan (Liu et al., 2013).
- In modelling the linkages between vulnerability factors, disaster impacts and losses, the data were collected from multiple sources. The disaster loss database regarding Typhoon Morakot had been systematically built by the Department of Science and Technology, Taiwan. This database included the surveyed numbers of casualties, property and agricultural losses, the distributions of inundation and landslides, and damaged public facilities. The data on vulnerability factors were obtained through combining official censuses and random sampling face-to-face questionnaire surveys to residents (shown in Table 1
- 15 in detail).

(Figure 2. Distributions of the estimated composite vulnerability indices over three river basins)

**4 Results and discussions**

**4.1 Composite vulnerability assessments**

Using the estimated CVIs by equation (3), Fig. 1 shows the distributions of estimated index values superimposed on the administrative boundaries of villages throughout the three river basins. The CVI estimates were divided into five levels (at 20% intervals). The villages with the estimated index values within the 80-100th percentiles can be defined as the most vulnerable, and those within the 1st-20th percentiles as the least vulnerable.

In Fig. 1, it shows that there are highly heterogeneous in the spatial distributions of the estimated composite vulnerability

註解 [H35]: Replies to the Comment [S36]: We have checked the language problem and reworded this sentence.

註解 [H36]: Replies to the Comment [S37]: We have replaced the term "geopolitical" with "administrative" in this sentence.

註解 [H37]: Replies to the Comment [S38]: We have checked the language problem and reworded this sentence. across the study areas. In the Tsengwen River basin, the most vulnerable areas concentrated in the middle reaches and some coastlines. Moreover, most of the middle and upper reaches of the Gaoping and Taimali River basin (especially northern shore) were distributed by the most vulnerable villages, while most of the lower reaches spread with the least vulnerable ones. These spatial distribution patterns conform to historical experience with which numeral typhoons had hit these areas in past years, and resulted in serious casualties, property and crop losses.

The results corroborate similar findings from related studies (Hung and Chen, 2013; Liu et al., 2013), asserting that significantly show that spatially-defined clusters of highly vulnerable areas are mostly situated in midstream and upstream reaches. This leads to a challenge for watershed managers in understanding of why these areas are particularly vulnerable and how they link to disaster losses, as well as what the implications of this might be for land-use planners to reduce risk.

**10 4.2 The distributions of losses due to Typhoon Morakot**

Morakot, a Category 1 typhoon, hit southern Taiwan during 8-12 August 2009. It was the most severely damaged typhoon in Taiwan in the past 50 years. This typhoon caused torrential rainfall that results in widespread flooding and thousands of landslides. Typhoon Morakot killed nearly 700 people and left thousands of people either displaced or homeless. The estimated total amount of economic losses was approximately US\$ 0.6 billion (Liu et al., 2013). The inundation areas due to

- 15 Typhoon Morakot were concentrated in the convergent regions of Gaoping River with its tributaries, while the major landslide and debris flow torrents occurred in the middle and upper reaches. This would affect the distributions of property, public facility and agricultural damage (Fig. 3). Using t test for correlation analysis, it shows that the location of agricultural damage significantly corresponded to where the landslides (Spearman  $\rho = 0.18$ , p

(1) Kaoping River basin

(3) Tsengwen River basin

Figure 1. Distributions of the estimated composite vulnerability indices over three river basins

---

## Author Comment (AC2) · 5 Sep 2016

Replies to the interactive comment on "Linking local vulnerability assessments to climatic hazard losses for river basin management" Anonymous Referee #2

1.The authors present a study that seeks to link typhoon losses within three river basins with vulnerability and coping capacity/adaptive capacity indicators. Particularly, the authors try to determine the statistical relationships between selected indicators and the observed losses using various regression methods. The presented study could provide interesting empirical/quantitative insights into the usefulness and the effects of, e.g., mitigation methods in the context of risk management, and might thus be considered for publication. There are, however, numerous substantial issues that need to

be addressed by the authors, primarily in regard to the underlying conceptual framework. *Replies: The authors would like to sincerely appreciate the anonymous referee for providing us very valuable suggestions and comments. Our responses and relevant ways of dealing with each comment and suggestion are summarized as follows. Particularly, we will further clarify our article's conceptual framework and rewrite major portions of the manuscript to make it more understandable.

2.In the following, I am listing my main concerns. Despite those, the manuscript requires substantial language editing. I have just listed selected instances where language editing is required. *Replies: Many thanks for referee's suggestions. The authors will follow the referee's suggestions to improve the writing and the language problem in the manuscript. We will reorganize whole manuscript, including the word choice, grammar, sentence usage and structure. Finally, the revised manuscript will also have a thorough language editing for making the content intelligible and making the manuscript more suitable for publication.

3.The authors state their aim to "examine whether geographic localities characterized by high vulnerability experience significantly more damage [. . .]" (p. 3, l. 5). Significantly more damage in comparison to what? Please rephrase and clarify. *Replies: The sentence will be rephrased as: "This article aims to examine whether localities characterized by high vulnerability experience significantly higher damage than other areas owing to onset weather extreme events at the river basin level. . ." in the revised manuscript.

4.It is argued that several events hit the studied river basins. Maybe provide some more examples on additional events to put the study into a broader context, since just a singular event, i.e., Morakot, surely would not trigger exhaustive risk management but might be more treated like an outlier. *Replies: Indeed, this is a critical limitation in our study. In the study, we adopt a single case of Typhoon Morakot due to two key reasons. First, Typhoon Morakot was the most severely damaged typhoon to hit Taiwan in the past 50 years. This typhoon caused torrential rainfall that resulted in widespread

flooding and thousands of landslides in the southern Taiwan areas. The damaged areas included all the three river basins discussed in the article, even though there were also several typhoons having hit southern Taiwan in the past years. This also leads to the following reasons for using only the Typhoon Morakot case. Second, except for Typhoon Morakot, most of the typhoons, which had hit southern Taiwan areas, are relatively scattered and smaller scales. This would cause difficulties in comparing disaster loss data between the various typhoons, while these typhoons are associated with different wind speeds, types of disaster impacts and various scales of precipitation and damaged areas. Particularly, most of these typhoons are deficient in the document of disaster losses and related database for supporting our analyses. Thus, our study focuses on developing a framework and methodology that allow us to link the local vulnerability assessments to climatic hazard losses at the river basin levels. We not only demonstrate the approach using Typhoon Morakot data as our initial step, but also recognize the potential of the application of our approach to other typhoons and other river basin units elsewhere in Taiwan and in the world. Indeed, the related discussions about these issues were really deficient in the original manuscript. Regarding the problems and limitations about the single-typhoon approach used in the study, we will add more discussions in the section 4.4 (Policy implications) in the revised manuscript.

5.P. 4, l. 2: I feel like you need more detail/consistency in your approach when it comes to the usage of terms, to better describe your conceptual framework (this is a comment also in light of the vague definition of vulnerability, please see a comment on that below). Here, what are biophysical elements, socioeconomic elements, industrial elements (aren't these belonging to socio*economic* elements?), "and land use elements" (sic!). Don't all these elements constitute some form of land-use? *Replies: In the manuscript, it really lacks detailed definition of the conceptual framework of vulnerability and some terms involved in our analyses. Thus, in the revised manuscript, the authors will further identify the key terms used in the analytical framework such as vulnerability and its components (e.g., exposure, biophysical context, socioeconomic sensitivity, land use and adaptive capacity). In particular, we will further clarify the process of building the conceptual framework of vulnerability, which would help us define the key components (and factors) of vulnerability.

6.P. 4, l. 5: What is meant with properties of a specific watershed context? *Replies: The properties of a specific watershed context mentioned here could mainly refer to hazard (for example, geology) attributes (they could also include ecology and institution contexts, but these properties would be beyond the scope of our discussion) other than vulnerability components. To clarify the wording of this sentence, we will replace "properties" with "hazard attributes" in this sentence.

7.The definition of vulnerability that is used by the authors remains vague. Please elaborate more on the concepts of exposure, sensitivity, adaptive capacity employed in your framework as shown in Eq. 1. I take it that, here, exposure refers to the UNISDR definition, i.e., elements potentially at risk? Also, later on, in Eq. 3, you refer to loss as a function of hazard and vulnerability. Thus, following your argumentation, e.g., I do not see averaged annual rainfall or debris flow as elements of the entity exposure (and, therefore, vulnerability), but rather as entities of the hazard (or hazard magnitude) used to determine affected (exposed) area/elements. I would also argue on the indicators elevation and proximity to rivers, which I do not see as an indicator for sensitivity. In order to clarify such issues, please elaborate in more detail on your conceptual framework/the definitions employed in your study. Also, if you argue that vulnerability frameworks need to be integrated, why didn't you investigate into additional, integrated frameworks of vulnerability? *Replies: The authors are so grateful for the referee's suggestions. In the manuscript, it really needs to elaborate more on the conceptual framework of vulnerability and its components. Therefore, in the revised version of the manuscript, we not only will increase more descriptions about how the conceptual framework of vulnerability was built, but also will add more explanations about the definition of exposure, sensitivity and adaptive capacity. First, we increase the discussions about the conceptual framework of vulnerability that was developed based on combining the vulnerability concept of IPCC (2014) with the risk-hazard approach (UNISDR,

2013). This leads to conceptualize vulnerability as a function of exposure, sensitivity and adaptive capacity. Second, according to a combination of the concepts of IPCC and risk-hazard approach, we also define the components of vulnerability using a hybrid concept from these two approaches. We define exposure as "the presence of areas, system or assets in places and settings that could be adversely affected." (IPCC, 2014). Thus, exposure can be considered as the nature and degree to which an area is exposed to climatic hazards. Moreover, sensitivity can be defined as "the degree to which a system is affected by climatic-related hazards" (IPCC, 2014). Based on these concepts, in our manuscript, "rainfall" and "debris flow torrents" are measured by long-term trends of an area exposed to flood hazard and debris flow torrents. However, as the referee's comment, it is really better for considering the "biophysical context" indicators as a component of "exposure" than as an element of "sensitivity". Thus, in the section 2 (particularly, 2.1 Vulnerability assessments) and section 3 (particularly, 3.1 Indicators of the vulnerability framework and hypotheses) of the revised manuscript, the authors will adjust "biophysical context" indicators from "sensitivity" dimension to "exposure" dimension in order to suit their definitions. The authors agree with the referee's suggestions, so we use an indicator-based integrated approach to aggregate various components of vulnerability. Based on our hypothetical relationships between various indicators and vulnerability, we apply a multicriteria decision analysis (MCDA) procedure to aggregate various indicators to calculate the composite vulnerability index (CVI), which can represent the integrated vulnerability of each spatial unit (a village). In the section 3 (Methods and data) of revised manuscript, we will increase a figure (Figue 1) to help understand the procedure of assessing integrated vulnerability and to describe the stepwise procedure for linking local vulnerability assessments to hazard loss analyses.

8.In doing so, please also provide more insight into why the criteria listed in table 1 have been selected/how selection has been done. *Replies: The criteria used in our study are chosen based on our conceptual framework of vulnerability, literature review and the contextual characteristics of the river basin management. Especially, our definition

of vulnerability and the selection of indicators refer to Hung and Chen (2013), which has been widely applied in the river basin vulnerability assessments in Taiwan. Thus, some processes of indicator selection were discussed in detail in Hung and Chen (2013). In the section 3 (Methods and data) of the revised manuscript, the authors will increase more descriptions to clarify the process of criteria and indicator selection.

9.P. 7, l. 18: It could be argued if increasing income etc. enhances coping strategies so that vulnerability is reduced, since an increase or accumulation of wealth etc. is typically also seen to increase vulnerability by increasing the values potentially at risk. *Replies: We would agree with referee's comments on the relationship between income and vulnerability. It is a reasonable relation between income and vulnerability concept. In fact, there is no common view about the relationship between income and vulnerability concept in the existing studies. In our study, we focus on a hypothesis of increasing income can be expected to enhance coping strategies. However, it needs to be examined by the case study. Finally, our case study showed that the areas with higher income would have lower typhoon losses than the other areas. But it is still in need of more case studies to reinforce the findings about the relationship between income, vulnerability concept and disaster losses.

10.P. 7, l. 23: If you argue that vulnerability is a multi-dimensional concept (which is not defined in the manuscript), I do not see how a preservation of (environmentally) sensitive (i.e., vulnerable?) areas leads to a decrease in vulnerability? *Replies: In the section 2.2 (Vulnerability and disaster losses) of the revised manuscript, the authors will define the multi-dimensional concept of vulnerability as that the components of vulnerability include various biophysical and socioeconomic factors of a specific river basin area. Moreover, we will also increase more explanations about how a preservation of environmentally sensitive areas leads to a decrease in vulnerability. In the study, the expectation is that a preservation of more environmentally sensitive areas would reduce urban, agricultural, and road developments, which could result in a decrease in vulnerability and lead to less disaster losses.

11. P. 6, l. 12: What are the synergies and complexities that have been obtained and discussed by Hung & Chen? What is their relevance for the presented study? What is the "framework of vulnerability indicators" that is being referred to? As mentioned above, please elaborate more on your methodology and conceptual frame. *Replies: Many thanks for the referee's suggestions. In the section 3.1 (Indicators of the vulnerability framework and hypotheses) of the revised manuscript, we will add more explanations about the relations between our study and Hung and Chen's (2013) approach. Especially, we will elaborate more on the process of building the conceptual framework of vulnerability and the stepwise procedure for linking local vulnerability assessments to hazard loss analyses (by adding a figure to supporting the explanations).

12. In regard to Eq. 3, it is argued that using just one event (i.e., the typhoon Morakot) allows for the control of the "disaster scenario", so that variation in loss can be attributed to variation in vulnerability. How do you control for hazard intensity as a governing factor of loss? Again, please provide more information on the methodology. *Replies: Indeed, there are some confusing descriptions in this sentence. Thus, it needs more explanations about the equation (3) to clarify its implications. In the revised manuscript, the sentence: "this approach allows disaster scenarios to be controlled, and which, other things being equal, any variation in losses directly are resulted from changes in hazard impacts and vulnerability factors..." will be reworded as "This approach allows us to concentrate on single disaster scenario, so that any variation in losses can be directly resulted from changes in hazard impacts and vulnerability factors...". In addition, we will also add more explanations (by using a figure) about the procedure and methodology used in our study.

13. I have the impression that also the term hazard impact needs more clarification. Following table 1, hazard impact seems to refer to casualties, losses etc. However, in the manuscript, p. 9, l. 13, it is argued that the interaction of hazard impacts and vulnerability generates loss. In this case, wouldn't impact refer to hazard intensity? *Replies: Indeed, some further clarification of the equation (3) is needed. Based on

the concept of risk-hazard approach, we assume that disaster risk/loss is a function of hazard and vulnerability. In a long-term scale, it implies that hazard risk is the product of hazard and vulnerability. However, in a short-term or single hazard event scale within a specific region, it is a deterministic approach, which hazard can be varied with its different intensities and impacts on various areas. In our study, because we adopted a single hazard event approach, the hazard impact on various areas will be used for measuring "hazard" in the equation (3). This implies that the term "hazard impacts" has involved "hazard intensity" concept. The related explanations in the original manuscript are not clear enough. Thus, in the revised manuscript, we will reinforce the related discussions about the term "hazard" in the equation (3).

14. In table 1, you list various indicators per category. Please, describe what is the Mean/standard deviation referring to? Is, e.g., mean of population density equal to 2.74 inhabitants/km2/village unit? Isn't this number rather low, also considering your argumentation that the three river basins have approximately 1.26 mio. Inhabitants and an area of 7885km2 (which would equate to approximately 160 inh./km2)? *Replies: The authors would like to appreciate the referee's reminders. In the revised manuscript, we will add a note to explain that the mean and S.D. are average and standard deviation values of the villages. On the other hand, the unit of "mean of population" is "thousand people" rather than the number of people. These revisions will be made in the revised manuscript.

15. Please provide more information on your indicators used to assess adaptive capacity. E.g., you mention the indicator "access to resources". What is meant with this indicator? What resources, access in which way, etc., could you please provide more detail? How is the indicator operationalized? Also in this regard, what is the number of interviews in total, and what is N per river basin? *Replies: The interview data of the indicators used to assess adaptive capacity, including "risk perceptions", "access to resources" and "adaptation appraisal", were collected from a questionnaire survey of random sampling of the households for each village. This questionnaire survey

was conducted by the National Science and Technology Center for Disaster Reduction (NCDR), Taiwan in 2004. The survey mainly aims to understand the extent of residents' hazard (e.g., flood, debris flow torrents) risk perceptions, hazard reduction and adaptive behavior as well as their determinants. The surveyed area included entire Taiwan, and the total sample size was 2,913. The related explanations about the questionnaire survey data was really deficient in the original manuscript. In the section 3 (Methods and data) and Table 1 of the revised manuscript, the authors will add more descriptions about these data and their sources.

16.Please elaborate more on the approaches that you classify as "top-down" and "bottom-up". *Replies: In the section 2 (Vulnerability and disaster impacts) of the revised manuscript, the authors will increase more explanations about the differences between "top-down" and "bottom-up" approaches for linking vulnerability and disaster losses. In our study, we identify the "top-down" approaches focus on combining existing hazard loss theories (such as risk-hazard, PAR, PSR theory or MCDA) with computer-aid simulation and GIS-based analysis to project disaster losses. The "bottom-up" approaches concentrate on characterizing the distributions of disaster damage using historical or survey databases. However, these two types of approaches are often applied interactively rather than independently in the disaster damage (or risk) analyses. The related discussions were really rare in the original manuscript. Thus, in the revised manuscript, we will reinforce the related explanations in the section 2 (Vulnerability and disaster impacts).

17.Also, in regard to both approaches, you argue on the importance of connecting them, and on their "relative magnitude" (p. 5, l. 21). What is meant with that, what is relative magnitude? Furthermore, do you see both types of approaches as distinct, since you argue for a further integration of both? Isn't it the case, however, that "top-down" approaches also make use of empirical data (losses etc.) for validation of models, hence, integrate both approaches? *Replies: In the sentence "Increasing the understanding of the evolution of climatic disaster risk highlights the importance of

connecting these two approaches and their relative magnitudes". It is really not clear enough. Thus, in the revised manuscript, the authors will reword this sentence as "Increasing the understanding of the formation of climatic disaster risk highlights the importance of connecting aforementioned two types of approaches and their relative magnitudes in hazard risk analyses". This would provide a clearer description. Really, as above mentioned, the distinction between the "top-down" and "bottom-up" approach is not absolutely clear and often used interactively. Our study aims to provide an approach to link these two types of approaches. Therefore, the terms "top-down" and "bottom-up" in the manuscript only provide a rough classification method for existing studies, which can help us identify the related contributions of our study. Finally, we will elaborate more on our approaches and related terms used in the manuscript.

18.Generally, please provide more details on your statistical analysis. I suggest you to include e.g. results as supplementary material if possible, e.g., correlation coefficients, etc. I assume that your analysis is carried out per river basin and per village unit, i.e., non-spatial? *Replies: In the revised manuscript, the authors will provide some supplementary materials, such as correlation coefficients, to help readers understand the results of some statistical analyses. Moreover, our analysis is conducted by per river basin (including entire three river basins) and per village unit. The spatial statistic analysis is majorly applied in the vulnerability assessments by using an MCDA procedure.

19.I find it difficult to interpret Spearman/Pearson r without knowledge on the shape of the distribution of indicators _ losses, as e.g. Pearson would only make sense in linear relationships. Do you find linearity? Again, what is N per river basin? *Replies: In reality, we are not very sure about whether or not the relationships between various disaster losses are linear. We assumed that there are linear relationships between these disaster loss items in order to simplify our analyses, although there are alternative methods to examine their relationships. Furthermore, we are also not really sure the distributions of disaster loss data are normal. Therefore, we simultaneously used Person correlation coefficient and nonparametric statistics (Spearman correlation coefficient) to test their correlations. In the revised manuscript, we will increase more explanations about these assumptions.

20. Abstract, l. 8: Please rephrase "attack". *Replies: We will reword and rephrase this sentence.

21. Introduction, P. 1, l. 24: Remove meanwhile. It sounds like 700 people were killed during the typhoon due to something completely different. *Replies: We will remove "meanwhile" and reword this sentence.

22. (*) P. 5, ll. 12-18: The paragraph is unclear to me. *Replies: The authors will reword and rephrase this paragraph to more clarify its discussions.

23. P. 10, l. 1.: Please rephrase, use e.g. people or inhabitants instead of populations. *Replies: We will replace "populations" with "inhabitants" in this sentence.

24. P. 10, l. 18: C4 I guess you mean administrative instead of geopolitical boundaries? *Replies: We will also replace "geopolitical" boundaries with "administrative" boundaries.

25. P. 11, l. 9: Do you mean spatially defined clusters? *Replies: This sentence will be reworded as "spatially-defined clusters of highly. . .".

26. P. 12, l. 13: 93% of typhoon Morakot caused no damage or injuries. This sounds a bit odd. *Replies: This situation would result from our applying single typhoon event to characterize disaster loss distributions. It also causes us to use ZIP (zero-inflated Poisson) and ZINB (zero-inflated negative binomial) regression model to examine the determinants of disaster casualties. If more typhoon cases are considered, the distributions of typhoon injuries or fatalities would be more even. We will add more sentences about this situation in the discussions of our case study findings.

[Figure]

**Figure 1.** Stepwise procedure and framework of analysis